# The Susceptibility of Retinal Ganglion Cells to Optic Nerve Injury is Type Specific

**DOI:** 10.3390/cells9030677

**Published:** 2020-03-10

**Authors:** Ning Yang, Brent K Young, Ping Wang, Ning Tian

**Affiliations:** 1VA Salt Lake City Health Care System, Salt Lake City, UT 84148, USA; yangning0903@hotmail.com (N.Y.); brent.k.young@utah.edu (B.K.Y.); ping.wang@utah.edu (P.W.); 2Ophthalmology and Visual Sciences, University of Utah, Salt Lake City, UT 84132, USA; 3Interdepartmental Neuroscience Program, University of Utah, Salt Lake City, UT 84114, USA

**Keywords:** retinal ganglion cell death, optic nerve crush, NMDA excitotoxicity, RGC type-specific susceptibility

## Abstract

Retinal ganglion cell (RGC) death occurs in many eye diseases, such as glaucoma and traumatic optic neuropathy (TON). Increasing evidence suggests that the susceptibility of RGCs varies to different diseases in an RGC type-dependent manner. We previously showed that the susceptibility of several genetically identified RGC types to *N*-methyl-D-aspartate (NMDA) excitotoxicity differs significantly. In this study, we characterize the susceptibility of the same RGC types to optic nerve crush (ONC). We show that the susceptibility of these RGC types to ONC varies significantly, in which BD-RGCs are the most resistant RGC type while W3-RGCs are the most sensitive cells to ONC. We also show that the survival rates of BD-RGCs and J-RGCs after ONC are significantly higher than their survival rates after NMDA excitotoxicity. These results are consistent with the conclusion that the susceptibility of RGCs to ONC varies in an RGC type-dependent manner. Further, the susceptibilities of the same types of RGCs to ONC and NMDA excitotoxicity are significantly different. These are valuable insights for understanding of the selective susceptibility of RGCs to various pathological insults and the development of a strategy to protect RGCs from death in disease conditions.

## 1. Introduction

In mammals, retinal ganglion cells (RGCs) are the only output neurons that conduct visual signals from the eyes to the brain. RGCs are classified into at least 40 types based on their morphological, functional and genetic features [1,2,3,4,5,6,7,8,9]. RGC death occurs in many blinding retinal diseases, such as glaucoma and traumatic optic neuropathy (TON). Increasing evidence suggests that RGCs are susceptible to various injuries in an RGC type-dependent manner. For instance, in experimental models of ocular hypertension, OFF RGCs exhibit higher rates of cell death than ON RGCs [10,11,12], and mono-laminated ON RGCs are found to be more susceptible to elevated intraocular pressure (IOP) than bi-laminated ON–OFF cells [13]. Similarly, in models of optic nerve injury, OFF RGCs are more susceptible than ON RGCs, and ON-sustained RGCs seem to be more vulnerable than ON-transient RGCs [14]. Among several types of RGCs, αRGCs are the least susceptible RGC type to an optic nerve injury in one report [15] but a more susceptible RGC type in another study [16]. It was shown that different RGC types have unique gene expression patterns [7,9,17,18], and the same genes could protect some RGC types but facilitate the death of other RGC types after the same injury [19]. Therefore, an understanding of the type-specific susceptibility of RGCs may provide valuable insights into the molecular mechanisms of RGC death and suggest novel treatment strategies.

Recent studies have provided valuable information regarding the molecular mechanisms of RGC death in retinal diseases and suggested multiple mechanistic pathogenesis processes. For instance, it was proposed that glaucomatous damage is a result of elevated intraocular pressure (IOP) followed by ischemia, hypoxia of the optic nerve head, and consequently, RGC death due to glutamate-induced excitotoxicity, deprivation of energy and oxygen, an increase in levels of inflammation and alteration of the flow of trophic factors. These events lead to blockage of anterograde and retrograde axonal transport with ensuing axotomy and eventually blindness [20,21,22,23,24,25,26]. Similar to glaucoma, the precise mechanisms of RGC death in TON have not been elucidated, but the pathology appears to be multifactorial, and several mechanisms of RGC death have been postulated, such as axonal transport failure, neurotrophic factor deprivation, activation of apoptotic signals, mitochondrial dysfunction, excitotoxic damage, oxidative stress, misbehaving reactive glia and loss of synaptic connectivity [26,27,28,29]. Glutamate excitotoxicity plays an essential role in RGC death of both glaucoma and TON. For instance, elevated intraocular pressure (IOP) increases the expression of *N*-methyl-d-aspartate receptors (NMDARs) in DBA/2J mice (homozygous for *Cdh23^ahl^*, Stock No. 000664, Jackson Lab) [30], and activates NMDARs, which in turn triggers mitochondria-mediated apoptotic cell death in glaucomatous retina [31]. Further, the numbers of NMDAR-positive RGCs are reduced parallel to the loss of RGC in a rat chronic ocular hypertension model [32]. However, to what extent NMDA excitotoxicity or direct mechanic crush causes the death of various types of RGCs has not been systematically investigated.

We previously showed that the susceptibility of twenty RGC types to NMDA excitotoxicity varies significantly [31]. One of these mouse lines expresses YFP in a direction-selective RGC type (BD-RGC). BD-RGCs are a type of ON–OFF direction-selective RGCs (DS-RGCs). In mouse retinas, there are three types of ON–OFF DS-RGCs, tuned to ventral, dorsal, nasal, and temporal motion. BD-RGCs are sensitive to ventral motion [33,34]. The second mouse line expresses YFP in W3-RGCs. W3-RGCs are the smallest RGCs in the size of the dendritic field and the most numerous RGCs [33]. There are at least two subtypes of W3-RGCs. W3B RGCs are ON–OFF motion-sensitive RGCs [33], and W3D RGCs remain physiologically uncharacterized [35,36]. Both W3-RGC subtypes express YFP in this mouse line. The third mouse line expresses YFP in αRGCs. There are at least three subtypes of αRGCs in mouse retinas, in which some are ON cells and some are OFF cells [37,38]. In this mouse line, YFP is expressed in all three subtypes of αRGCs [15]. The fourth line of the transgenic mice expresses YFP in J-RGCs. There are three subtypes of J-RGCs in the mouse retina, which differ in dendritic tree morphology and possibly visual function [3,7,33]. This mouse line expresses YFP in two subtypes of J-RGCs. One subtype of J-RGCs orients its dendrites ventrally to form a polarized dendritic field and is sensitive to directional movement, color-opponent responses, and orientation-selective response [3,33,39,40]. The second subtype of J-RGCs has a symmetric dendritic field, and the function of them is not well characterized [3]. The final mouse line expresses YFP in 12 morphological types of RGCs [41]. Because it has been widely reported that the susceptibility of RGCs varies based on their dendritic ramification patterns, type of their light responses, and their genetic profiles [7,9,10,11,13,14,15,17,18], we used these transgenic mouse lines that provided multiple RGC types with various morphological, physiological and genetic profiles.

In this study, we characterized the susceptibility of the same RGC types as those for NMDA excitotoxicity to optic nerve crush (ONC). We show that the susceptibility of different types of RGCs to ONC varies significantly, in which BD-RGCs are the most resistant type of RGCs while theW3-RGCs are the most sensitive cells to ONC. Further, our results show that the survival rates of BD-RGCs and J-RGCs after ONC are significantly higher than the survival rates of BD-RGCs and J-RGCs after NMDA excitotoxicity. These results provide valuable insights for understanding the selective susceptibility of RGCs to pathological insults and the development of a strategy to protect RGCs from death in disease conditions.

## 2. Materials and Methods

### 2.1. Animals

The transgenic mouse strains used in this study are the same as in our previous study [31], which include B6.Cg-Tg(Thy1-YFP)HJrs/J (Thy1-YFP), B6.129(SJL)-Kcng4tm1.1(cre)Jrs/J (Kcng4^Cre^), FSTL4-CreER (BD-CreER), JamB-CreER, TYW3, and Thy1-STOP-loxP-YFP (Thy1-Stop-YFP). The Thy1-YFP (Stock No: 003782) and Kcng4^Cre^ (Stock No: 029414) mice were obtained from The Jackson Laboratory (Bar Harbor, ME, USA) [15]. BD-CreER, JamB-CreER, TYW3, and Thy1-Stop-YFP mice were obtained from Dr. Joshua Sanes’ laboratory at Harvard University [3,33]. All transgenic mice used in this study were on C57BL/6 background and were backcrossed with C57BL/6J mice for 4–5 generations in our lab. Then the BD-CreER, JamB-CreER, and Kcng4^Cre^ mice were bred into the Thy1-Stop-YFP mice to generate BD-CreER:Thy1-Stop-YFP (BD:YFP), JamB-CreER:Thy1-Stop-YFP (JamB:YFP) and Kcng4^Cre^:Thy1-Stop-YFP (Kcng4^Cre^:YFP) double transgenic mice. YFP was explicitly expressed in αRGCs without any additional treatment, whereas YFP was only expressed specifically in BD-RGCs or J-RGCs upon intraperitoneal (IP) injection of Tamoxifen (150 μg) at the ages of postnatal day 5-15 (P5–15). All of these mice were viable, and no significant defects in general development or overall formation of eye or retina were noticed. Mice in both sexes were used in this study. All animal procedures used in this study and care were performed following protocols approved by the IACUC of the University of Utah (Protocol number: 17-05007) and the IACUC of VA Salt Lake City Health Care System (Protocol number: A15-04) in compliance with PHS guidelines and with those prescribed by the Association for Research in Vision and Ophthalmology (ARVO).

### 2.2. Optic Nerve Crush

The ONC procedure has been described in our previous publication [42] and was performed unilaterally on all mice at approximately the age of P60–90. Isoflurane (2–5%, MWI, Meridian, ID) was used through a computerized mouse anesthesia suite (SomnoSuite1 System, Kent Scientific Corporation, Torrington, CT, USA) to anesthetize the mice. A topical application of 0.5% proparacaine hydrochloride ophthalmic solution (Falcon Pharmaceuticals, Fort Worth, TX, USA) was also used. Under a stereo surgical microscope, a small cut was made at the lateral canthus of the eyelid to expose the lateral side of the eyeball. Then, a small incision was made in the conjunctiva beginning inferior to the eyeball and around the cornea temporally. Microforceps were used to hold the edge of the conjunctiva next to the eyeball and retract it. The orbital muscles were gently deflected and the eyeball was rotated nasally to exposes the posterior aspect of the eyeball and optic nerve. Dumont #N7 cross-action forceps (cat. #RS-5027; Roboz) were used to hold the optic nerve at approximately 1 mm from the back of eyeball for 10–30 s (for different experiments, see results), with only pressure from the self-clamping action of the forceps to press on the nerve. The Dumont cross-action forceps have a spring action, which applies a constant and consistent force to the optic nerve. After 10–30 s, the optic nerve was released, and the forceps were removed to allow the eyeball to rotate back into place. A small amount of surgical lubricant (KY jelly; McNeil-PPC, Skillman, NJ, USA) was applied to the eye to protect it from drying, and a subcutaneous injection of buprenorphine was administered for post-operative pain control. The mouse was placed on a warming pad and monitored until it fully recovered from anesthesia. During the first three days after the procedure, systemic analgesics (buprenorphine) and topical antibiotic ointment were applied twice daily. The mice were closely monitored for possible infection, bleeding, and loss of muscular control [43,44,45]. The effectiveness of the injury of RGC axons was confirmed by intraocular injection of Alexa Fluor™ 555-conjugated Cholera Toxin Subunit B (CTB, 0.2%, ThermoFisher Scientific, Eugene, OR, USA) to label the optic nerve one week after the ONC.

### 2.3. Primary Antibodies

Rabbit polyclonal antibody against the green fluorescent protein (GFP) conjugated with AlexaFluor 488 was purchased from Molecular Probes (Eugene, OR, USA; Catalog No. A21311). This antibody was raised against GFP isolated directly from *Aequorea victoria* and has been previously characterized by immunocytochemistry in granule cells [46], olfactory sensory neurons [47], and hippocampal neurons that express GFP [48]. An anti-RBPMS (RNA-binding protein with multiple splicing) antibody was purchased from PhosphoSolutions (Aurora, CO, USA; Catalog #: 1832-RBPMS). This polyclonal antibody was raised in guinea pigs against a synthetic peptide corresponding to amino acid residues from the N-terminal region of the rat RBPMS sequence conjugated to KLH. This antibody has been characterized by Western blotting and verified with immunocytochemistry on mammalian retinas and demonstrated to be a specific pan-RGC marker which labels all RGCs but not any other cells in the retina [49,50,51]. The secondary antibodies were purchased from Jackson Immune Research Laboratories (West Grove, PA, USA).

### 2.4. Preparation of Retinal Whole Mounts for Antibody Staining

Retinal ganglion cells were imaged on whole-mount retinal preparation for cell counting and dendritic morphology recognition. The procedures for fluorescent immunolabeling of YFP-expressing retinal neurons on retinal whole mounts and slide preparations have been described previously in detail [31,41,42]. In brief, mice were euthanized with 100% CO_2_, followed by cervical dislocation. Retinas were isolated and fixed in 4% paraformaldehyde (PFA) in 0.01M phosphate-buffered saline (PBS; pH 7.4) for 30 min at room temperature. Fixed retinas were washed 10 min × 3 in 0.01M PBS and incubated in a blocking solution (10% normal donkey serum) at 4 °C for two h. Next, retinas were incubated in a guinea pig polyclonal anti-RBPMS antibody (1:500) and a rabbit polyclonal anti-GFP antibody conjugated with Alexa Fluor488 (1:500) for seven days at 4 °C to label the total RGCs and the YFP-expressing RGCs, respectively. A Cyanine Cy^TM^ 3-conjugated donkey anti-guinea pig (1:400, Jackson ImmunoResearch, West Grove, PA, USA) secondary antibody was used overnight at 4 °C to reveal anti-RBPMS antibody staining. After the antibody incubation, the retinas were washed 3 × 10 min, and flat-mounted on Super-Frost slides (Fisher Scientific, Pittsburgh, PA, USA) with Vectashield mounting medium for fluorescence (Vector Laboratories, Burlingame, CA, USA).

### 2.5. Confocal Laser Scanning Microscopy and Image Sampling

Fluorescent images of fixed retinal tissue were collected with a dual-channel Zeiss confocal microscope (Carl Zeiss AG, Germany) with a C-Apochromat 40 × 1.2 W Korr water immersion lens. Image stacks of YFP-expressing RGCs in whole-mount retinas were collected at intervals of 0.5 μm. Imaris software (Bitplane, Inc., Concord, MA, USA) was used to align the multistacks of images together and adjust the intensity and contrast of images.

For image sampling, we use two different strategies for retinas with low or high densities of YFP-expressing RGCs to avoid potential bias of data sampling when the persons carrying out the histological analysis were not blinded to the treatment. For Thy1-YFP, BD:YFP and JamB:YFP mice, the YFP is expressed in a relatively low density of RGCs, and the expression level varies significantly among mice (from several to several hundreds of YFP-expressing RGCs per retina) but not significantly between left and right eyes [31]. Therefore, we imaged the whole retina and counted every YFP-expressing RGCs in the GCL layer of these mice. Mice were only excluded from data analysis when the total number of YFP-expressing RGCs in the whole retina of the control eye was less than 10 in order to avoid the results being skewed by mice with an extremely low number of YFP-expressing RGCs. For TYW3 and Kcng4^Cre^:YFP mice, which constitutively express YFP in all W3-RGCs and αRGCs, the density of YFP^+^ RGCs is very high [31] and the expression level does not vary significantly among mice or between left and right eyes [31]. We included every mouse assigned to this study for data analysis without exclusion. For image sampling, we scanned four squares (304 μm × 304 μm each) at four quarters of the retina, 600 μm away from the center of optic nerve head. The density of YFP-expressing W3-RGCs and αRGCs of each retina was averaged from the four squares. In addition to BD-RGCs and J-RGCs, BD:YFP mice and JamB:YFP mice also express YFP in a small fraction of amacrine cells located in the inner plexiform layer (INL) [33] but not displaced amacrine cells in the ganglion cell layer (GCL). However, Kcng4^Cre^:YFP mice express YFP in αRGCs and some bipolar cells [15]. In this study, we only included the YFP-expressing cells in the GCL of these mice.

### 2.6. Statistical Analysis

Data are all presented as the mean ± SE in the text and figures (Igor Pro, WaveMetrics, Inc., Lake Oswego, OR, USA). Student *t*-tests are used to examine the difference between two means (Excel, Microsoft, Redmond, WA, USA), and ANOVA tests are used to examine the difference among more than two means (Statview, Abacus Concepts, Berkeley, CA, USA).

## 3. Results

### 3.1. The Effects of Crush Time on RGC Death and the Time Course of RGC Death

ONC is a commonly used model for RGC injury due to TON or glaucoma. Although most of the experiments were performed using similar cross-action forceps, which apply a constant and consistent force to press on the optic nerve by the pressure from the self-clamping action of the forceps, different studies clamped the optic nerve for a different length of time (from 3 s to 60 s) [52,53,54,55,56,57,58]. To determine the potential variation by the ONC procedure on RGC death, we first tested whether various durations of clamping cause RGCs to die to a different extent. Using the method established in our previous study and cross-action forceps [42], we clamped the optic nerve of the left eyes of wild-type mice for 10, 20, and 30 s. We collected the retinas of both the injured left eyes and the uninjured right eyes seven days after the ONC procedure. The RGCs of both crushed and uncrushed retinas were labeled using an anti-RBPMS antibody (Figure 1A). The retinas were imaged using a confocal imaging system to quantify the anti-RBPMS antibody labeled RGCs in both the injured and uninjured retinas. The density of surviving RGCs from the injured eyes was normalized to their uninjured opposite right eyes. The results were plotted as a function of the duration of clamping. The effectiveness of the crush to completely block axonal transport of RGCs was validated by the intraocular injection of Alexa Fluor^TM^ 555-conjugated CTB labeling of the optic nerve one week after the ONC (Figure 1B). The results demonstrated that the ratios of surviving RGCs with ONC of 10, 20, and 30 s have very little difference, although the difference is statistically significant (Figure 1D). Therefore, the variation in clamping time during ONC seems to have minimal effect on the extent of RGC death.

It has been shown that RGCs start to die 1–3 days after ONC and reach the maximum level of cell death approximately 30 days after ONC [59]. Accordingly, we validated the time course of RGC death after ONC in our experiments. We crushed the optic nerve for 20 s and examined the density of surviving RGCs at six time points from 3 days after ONC to 90 days after ONC. Our results showed that the number of surviving RGCs was reduced to 85% ± 2.3% of the uninjured opposite eyes three days after ONC (*n* = 6, *p* = 0.001, paired *t*-test, Figure 1E). The survival rate of RGCs gradually reduced with time after ONC (Figure 1C) and reached 12.7% ± 0.7% of the uninjured opposite eyes 30 days after ONC (*n* = 5, *p* < 0.0001, paired *t*-test, Figure 1E). After 30 days of post-ONC, the number of surviving RGCs did not change dramatically. The average number of surviving RGCs remained at 10.2% ± 0.6% (*n* = 6, *p* < 0.0001, paired *t*-test, Figure 1E) of the uninjured opposite eyes 60–90 days after ONC (Figure 1E). Therefore, the time course of RGC death after ONC in our experiments is consistent with previous reports [59].

### 3.2. YFP-Expressing RGCs of YFP-H Mice are More Resistant to ONC Than Total RGCs

Thy1-YFP mice have been used extensively for studying RGC morphology, physiology, development, and degeneration [31,60,61,62,63,64,65,66,67,68]. We recently tested whether the death of YFP-expressing RGCs of these mice could represent the death of total RGCs induced by NMDA excitotoxicity. Our results show that the survival rate of YFP-expressing RGCs is significantly higher than that of anti-RBPMS antibody labeled RGCs [31]. To further determine whether Thy1-YFP mice can be a reliable model for studying overall RGC death due to ONC, we quantified the survival rates of YFP-expressing RGCs and total RGCs of the same Thy1-YFP mice after unilateral ONC. Because the numbers of YFP-expressing RGCs vary significantly among Thy1-YFP mice, while the numbers of YFP-expressing RGCs of the left and right eyes of the same mice are comparable [31], we first established a control group for the study. Figure 2A shows representative images of YFP-expressing RGCs in flat-mount retinas of the left (A1) and right (A2) eye of a Thy1-YFP mouse. Figure 2B shows the average densities of YFP-expressing RGCs of the left and right eyes of 9 Thy1-YFP mice. Although a paired *t*-test showed that the difference between the densities of YFP-expressing RGCs of left and right eyes is statistically insignificant (paired *t*-test, *p* = 0.12), the density of YFP-expressing RGCs varies significantly among these mice as we reported previously [31]. Figure 2C shows anti-RBPMS antibody labeled RGCs of flat-mount retinas of the left (C1) and right (C2) eye of the same Thy1-YFP mouse. Similarly, the densities of anti-RBPMS antibody labeled RGCs of the left and right eyes are statistically insignificant (paired *t*-test, *p* = 0.25, Figure 2D). Therefore, the numbers of YFP-expressing RGCs, as well as the anti-RBPMS antibody labeled RGCs, of the left and right eyes are comparable.

Next, we tested whether the death of YFP-expressing RGCs of these mice could represent the death of total RGCs after ONC by comparing the survival rates of YFP-expressing RGCs and the RGCs labeled by the anti-RBPMS antibody of the same retinas. Accordingly, we performed a 20 s crush of the optic nerve of the left eyes of Thy1-YFP mice and used the uncrushed right eyes as controls. We then quantified the numbers of YFP-expressing RGCs and RGCs labeled by the anti-RBPMS antibody and normalized the density of surviving RGCs of ONC-treated left eyes to that of uncrushed right eyes. Figure 2E shows images of Thy1-YFP retinas without ONC (E1, right eye) and seven days after ONC (E2, left eye, 20 s clamp) of the same mouse. The density of YFP-expressing RGCs in the retina with ONC is lower than that of the control eye. Quantitatively, the density of YFP-expressing RGCs in the retinas with ONC is reduced to 62.5% ± 3.4% (*n* = 8, paired *t*-test, *p* < 0.0001) of the uncrushed right eyes (Figure 2F). On the other hand, the density of the RGCs labeled by anti-RBPMS antibody after ONC (G2, left eye) seems to have a more significant reduction in comparison with the uncrushed right eye (G1, right eye) of the same mice. Quantitatively, the average density of RGCs labeled by anti-RBPMS antibody in retinas with ONC is reduced to 33.3% ± 0.6% (*n* = 8, paired *t*-test, *p* < 0.0001) of uncrushed right eyes (Figure 2H). Therefore, the survival rate of YFP-expressing RGCs (62.5%) is significantly higher than that of anti-RBPMS antibody labeled RGCs (33.3%) (Figure 2I, *n* = 8, paired *t*-test, *p* < 0.0001), which is similar to what we observed in RGC death by NMDA excitotoxicity [31]. Accordingly, using the survival rate of Thu1-YFP RGC of Thy1-YFP H line to predict total RGC survival rate would significantly underestimate RGC death. Then, we employed four RGC type-specific transgenic mouse lines to study type-specific RGC death due to ONC.

### 3.3. BD-RGCs are More Resistant to ONC Than the Total RGC Population

It was reported that RGCs are susceptible to optic nerve injury in an RGC type-dependent manner [10,11,12,13,14,15,16]. We recently showed that the susceptibility of four RGC types to NMDA excitotoxicity varies significantly, in which the αRGCs are the most resistant type of RGCs to NMDA excitotoxicity. At the same time, the J-RGCs are the most sensitive RGCs to NMDA excitotoxicity [31]. To determine whether these four RGC types have different susceptibility to ONC, we used BD-CreER: Thy1-Stop-YFP (BD:YFP), JamB-CreER:Thy1-Stop-YFP (JamB:YFP), Kcng4^Cre^:Thy1-Stop-YFP (Kcng4^Cre^:YFP), and TYW3 transgenic mice for this study. YFP was expressed in BD-RGCs, J-RGCs, αRGCs, and W3-RGCs in these mice.

Similar to Thy1-YFP mice, the numbers of YFP-expressing BD-RGCs vary significantly among the BD:YFP mice, while the numbers of YFP-expressing BD-RGCs of the left and right eyes of the same mice are comparable [31]. Figure 3A shows a representative image of YFP-expressing BD-RGCs in a flat-mount BD:YFP retina (A1) and a magnified view of the area in the dash-line box in panel A1 to show the anti-RBPMS-labeled RGCs (red) and YFP-expressing BD-RGCs (green) (A2). We compared the density of the YFP-expressing BD-RGCs of the left and right eyes of 6 BD:YFP mice. The results show that the difference in the densities of YFP-expressing BD-RGCs of the left and right eyes of these mice is statistically insignificant (*n* = 6, paired *t*-test, *p* = 0.4, Figure 3B). In contrast, the density of YFP-expressing BD-RGCs varies significantly between individual mice.

We then quantified the susceptibility of BD-RGCs to ONC. Accordingly, we performed unilateral optic nerve crush for 20 s on the left eyes of BD:YFP mice between the ages of P60 and P90. We collected the retinas seven days after ONC and compared the densities of YFP-expressing BD-RGCs and the anti-RBPMS-labeled RGCs of the injured left eye with the uninjured right eye of each mouse. Figure 3C1 shows a representative image of YFP-expressing BD-RGCs in a flat-mount BD:YFP retina seven days after ONC. Figure 3C2 shows a magnified view of the area in the dash-line box of panel C1 to show the anti-RBPMS-labeled RGCs (red) and YFP-expressing BD-RGCs (green). Quantitatively, the average density of YFP-expressing BD-RGCs in retinas with ONC is reduced to 58.9% ± 5.9% (*n* = 10, paired *t*-test, *p* = 0.006) of the uncrushed right eyes. In contrast, the average density of RGCs labeled by anti-RBPMS antibody in the same retinas is reduced to 33.7% ± 1.4% (*n* = 10, paired *t*-test, *p* < 0.0001) of the uncrushed right eyes (Figure 3D). Statistically, the survival rate of YFP-expressing BD-RGCs (58.9% ± 5.9%) is significantly higher than that of anti-RBPMS antibody labeled RGCs (33.7% ± 1.4%) (paired *t*-test, *p* = 0.0005).

### 3.4. αRGCs are More Resistant to ONC Than the Total RGC Population

αRGCs have been reported to be relatively resistant to ONC. However, the results from different studies seem to be inconsistent. For instance, αRGCs are the least susceptible RGC type to ONC in one report [15] but more susceptible RGC type in another report [16]. We estimate their susceptibility to ONC by comparing the survival rate of αRGCs with the survival rate of the total RGC population labeled by the anti-RBPMS antibody of the same retinas. Different from BD-RGCs in BD:YFP mice, the densities of YFP-expressing αRGCs in uninjured Kcng4^Cre^:YFP mice seem to be more consistent both between mice and between the left and right eyes (Figure 4A,B). The difference in the densities of YFP-expressing αRGCs of the left and right eyes of these mice is statistically insignificant (*n* = 7, paired *t*-test, *p* = 0.64, Figure 4B).

We then performed the same procedure on Kcng4^Cre^:YFP mice as that on BD-YFP mice and quantified the survival rate of YFP-expressing αRGCs and anti-RBPMS antibody labeled RGCs of the same eyes seven days after ONC. The survival rate of αRGCs is significantly higher than that of the anti-RBPMS antibody labeled RGCs of the same eyes (Figure 4C). Quantitatively, the average density of YFP-expressing αRGCs in retinas with ONC is reduced to 53.7% ± 2.8% (*n* = 10, paired *t*-test, *p* < 0.0001) of the uncrushed right eyes. In contrast, the average density of RGCs labeled by anti-RBPMS antibody in the same retinas is reduced to 31.9% ± 1.2% (*n* = 10, paired *t*-test, *p* < 0.0001) of the uncrushed right eyes (Figure 4D). Statistically, the survival rate of YFP-expressing αRGCs (53.7% ± 2.8%) is significantly higher than that of anti-RBPMS antibody labeled RGCs (31.9% ± 1.2%) (paired *t*-test, *p* < 0.0001).

### 3.5. The Susceptibility of J-RGCs to ONC is Similar to that of Total RGC Population

We then examined the susceptibility of J-RGCs to ONC using JamB:YFP mice. JamB:YFP mice express YFP in two types of J-RGCs with distinctive dendritic morphology, one with an asymmetric dendritic field and another with a more symmetric dendritic field [3,33]. In this study, we count the YFP-expressing J-RGCs from both types. Figure 5A shows a representative image of a JamB:YFP retina (A1) and a magnified view of the area in the dash-line box of panel A1 to show the dendritic morphology of the J-RGCs with asymmetric dendritic fields (A2). Similar to BD:YFP mice, the numbers of YFP-expressing J-RGCs vary significantly among the JamB:YFP mice while the numbers of YFP-expressing J-RGCs of the left and right eyes of the same mice are comparable (Figure 5B). Quantitatively, the difference in the densities of YFP-expressing J-RGCs of left and right eyes of uninjured mice is statistically insignificant (*n* = 6, paired *t*-test, *p* = 0.78).

J-RGCs seem to be much more sensitive to ONC than BD-RGCs and αRGCs. Figure 5C shows a representative image of a JamB:YFP retina seven days after ONC (C1) and a magnified view of the area in the dash-line box of panel C1 to show both the anti-RBPMS antibody labeled RGCs (red) and YFP-expressing J-RGCs (green) (C2). In this JamB:YFP retina, the number of YFP-expressing J-RGCs is significantly reduced in comparison with the uninjured eye of the same mouse (Figure 5A). Quantitatively, the average density of YFP-expressing J-RGCs in retinas with ONC is reduced to 31.9% ± 4.5% (*n* = 8, paired *t*-test, *p* < 0.012) of the uncrushed right eyes. In contrast, the average density of RGCs labeled by anti-RBPMS antibody in the same retinas is reduced to 31.6% ± 1.9% (*n* = 8, paired *t*-test, *p* < 0.0001) of the uncrushed right eyes (Figure 5D). Statistically, the difference between the survival rates of the YFP-expressing J-RGCs (31.9% ± 4.5%) and the RGCs labeled by the anti-RBPMS antibody (31.6% ± 1.9%) in the same retinas are not significant (Figure 5D, paired *t*-test, *p* = 0.94). Therefore, the susceptibility of J-RGCs to ONC is similar to that of the total RGC population.

### 3.6. W3-RGCs are More Sensitive to ONC

Finally, we examined the susceptibility of W3-RGCs to ONC using TYW3 mice [33]. Similar to Kcng4^Cre^:YFP mice, but different from BD:YFP and JamB: YFP mice, TYW3 mice are conditional but not inducible transgenic mice. Therefore, the densities of YFP-expressing W3-RGCs in these mice are more consistent between mice and between the left and right eyes. Figure 6A shows a representative image of YFP-expressing W3-RGCs in a flat-mount TYW3 retina (A1) and a magnified view of the area in one of the dash-line boxes in panel A1 to show the anti-RBPMS-labeled RGCs (red) and YFP-expressing W3-RGCs (green) (A2). Quantitatively, the difference in the densities of YFP-expressing W3-RGCs of the left and right eyes of these mice is statistically insignificant (*n* = 7, paired *t*-test, *p* = 0.96, Figure 6B).

We then performed a 20 second ONC unilaterally on the left eyes of TYW3 mice as we did for all other mice. We quantified the survival rate of YFP-expressing W3-RGCs, and anti-RBPMS antibody labeled RGCs of the same eyes seven days after ONC. Figure 6C shows the magnified images of the retina without ONC (C1) and the retina seven days after ONC (C2) of a TYW3 mouse. In the TYW3 retina after ONC (Figure 6C2), the number of both the YFP-expressing W3-RGCs (green) and anti-RBPMS antibody labeled RGCs (red) is significantly reduced in comparison with the uninjured eye of the same mouse (Figure 6C1). Quantitatively, the average density of YFP-expressing W3-RGCs in retinas with ONC is reduced to 25.9% ± 2% (*n* = 9, paired *t*-test, *p* < 0.0001) of the uncrushed right eyes. In contrast, the average density of RGCs labeled by anti-RBPMS antibody in the same retinas is reduced to 33.5% ± 0.8% (*n* = 9, paired *t*-test, *p* < 0.0001) of the uncrushed right eyes (Figure 6D). Statistically, the survival rates of the YFP-expressing W3-RGCs (25.9% ± 2%) are significantly lower than that of the RGCs labeled by anti-RBPMS antibody (33.5% ± 0.8%) (Figure 6D, paired *t*-test, *p* = 0.001). Therefore, the susceptibility of W3-RGCs to ONC is higher than that of the total RGC population.

### 3.7. YFP-Expressing RGCs of YFP-H Mice are More Resistant to ONC Than Total RGCs

The results described above clearly demonstrated that the susceptibility of the YFP-expressing RGCs in the five transgenic mouse lines to ONC varies significantly, in which BD-RGCs are the most resistant RGC type, and W3-RGCs are the most sensitive RGC type to ONC. In our previous study of the susceptibility of the same RGC types to NMDA excitotoxicity, we show that αRGCs are the most resistant RGC type and BD-RGCs and J-RGCs are the most sensitive RGC types to NMDA excitotoxicity [31]. These results suggest that not only the susceptibility of RGCs to retinal injury is RGC type specific, but the susceptibility of the same RGC type might also change significantly to different primary pathological insults.

To further test this possibility, we directly compared the survival rates of the YFP-expressing RGCs of the five transgenic mouse lines under the conditions in which both ONC and NMDA excitotoxicity induced a similar survival rate of the RGCs as labeled by the anti-RBPMS antibody. Figure 7 shows the survival rates of the RGCs labeled by anti-RBPMS antibody (total RGC), the YFP-expressing RGCs in Thy1-YFP mice (Thy1-RGC), and the YFP-expressing BD-RGCs, αRGC, W3-RGC and J-RGC after ONC or NMDA excitotoxicity. The survival rates for ONC are the results of 7 days after 20 s of optic nerve clamping from the current study. The survival rates for NMDA excitotoxicity were the results of 24 h after the intraocular injection of 2 μL NMDA solution at the concentration of 3.125 mmol/L from our recent study [31]. The injuries by ONC and NMDA excitotoxicity resulted in similar survival rates of RGCs labeled by the anti-RBPMS antibody. Quantitatively, the average densities of RGCs labeled by anti-RBPMS antibody seven days after ONC and 24 h after intraocular injection of NMDA are 32.8% ± 0.6% (*n* = 45) and 32.2% ± 1.7% (*n* = 12), respectively. Statistically, the difference between these two survival rates is not significant (Figure 7, total RGC, unpaired *t*-test, *p* = 0.65).

Although the overall levels of RGC death by ONC and NMDA excitotoxicity are the same under these conditions, the survival rates of the three RGCs types, especially BD-RGCs and J-RGCs, are dramatically different. Quantitatively, the average survival rates of BD-RGCs are 58.9% ± 5.9% (n = 10) 7 days after ONC and 1.8% ± 0.6% (n = 6) 24 h after intraocular injection of NMDA, respectively. The difference is highly significant (Figure 7, BD-RGC, unpaired *t*-test, p < 0.0001). Similarly, the average survival rates of W3-RGCs are 25.9% ± 2% (n = 9) 7 days after ONC and 18.8% ± 1% (n = 5) 24 h after intraocular injection of NMDA, respectively. These are also significant (Figure 7, W3-RGC, unpaired *t*-test, *p* = 0.025). In addition, the average survival rate of J-RGCs 7 days after ONC (31.9% ± 4.5%, n = 8) is significantly higher than that of 24 h after intraocular injection of NMDA (2.1% ± 0.6%, n = 5, Figure 7, J-RGC, unpaired *t*-test, *p* = 0.0003), while the average survival rate of αRGCs 7 days after ONC (53.7% ± 2.8%, n = 10) is not significantly different from that of 24 h after intraocular injection of NMDA (47.4% ± 0.8%, n = 5, Figure 7, αRGCs, unpaired *t*-test, *p* = 0.142). These results demonstrated that not only the susceptibility of different RGC types to the same pathological insult varies significantly, the susceptibility of the same RGC types to different pathological insults also varies significantly. Therefore, different primary pathological insults might preferentially damage particular RGC types, which might result in specific functional defects and requires different treatment strategies. Furthermore, four out of five group/types of RGCs have higher survival rates 7 days after ONC than 1 day after NMDA injection although the survival rates of RGCs labeled by anti-RBPMS antibody are not significantly different under these two conditions. These results indicate that ONC seems to be a less damaging stress than NMDA excitotoxicity to these four group/types of RGCs. These results also imply that ONC has to be a more damaging stress than NMDA excitotoxicity to other RGC types in order to maintain a similar level of overall RGC death.

## 4. Discussion

Our results show that the susceptibility of different types of genetically identified RGCs to ONC varies significantly. Among the RGC types tested, BD-RGCs are the most resistant RGC type to ONC, while W3-RGCs are the more sensitive cell types to ONC. On average, the susceptibilities of both BD-RGCs and αRGCs are significantly lower than the susceptibility of the entire RGC population; the susceptibility of J-RGCs is not different from the susceptibility of the whole RGC population. In contrast, the susceptibility of W3-RGCs is slightly higher than the susceptibility of the entire RGC population. We also show that the survival rates of BD-RGCs, J-RGCs and W3-RGCs after ONC are significantly higher than their survival rates after NMDA excitotoxicity. These results strongly suggest that the differences in the genetic background of RGC types might provide valuable insights for the understanding of the RGC type-specific susceptibility and selective susceptibility of RGCs to different pathological insults. These results could also assist in the development of strategies for protecting RGCs under various disease conditions.

### 4.1. Classification of RGC Types

In mammals, RGCs are classified into at least 40 types based on their morphological, functional and genetic features [1,2,3,4,5,6,7,8,9]. This study includes five groups of RGCs with unique structural, functional, and genetic features. BD-RGCs are a type of ON–OFF direction-selective RGCs (DS-RGCs). In mouse retinas, there are three types of ON–OFF DS-RGCs, tuned to ventral, dorsal, nasal, and temporal motion. BD-RGCs are sensitive to ventral motion [33,34]. W3-RGCs are the smallest RGCs in the size of the dendritic field and the most numerous RGCs [33]. There are at least two subtypes of W3-RGCs. W3B W3-RGCs are motion sensitive, and W3D W3-RGCs remain physiologically uncharacterized [35,36]. There are at least three subtypes of αRGCs in mouse retinas [37,38]. Kcng4^Cre^:YFP mice express YFP in all three subtypes of αRGCs, and some subsets of bipolar cells [15]. There are three subtypes of JamB-expressing RGCs in the mouse retina, which differ in dendritic tree morphology and possibly visual function [3,7,33]. The JamB:YFP mice express YFP in two subtypes of JamB-expressing RGCs (J-RGCs). One subtype of J-RGCs orients its dendrites ventrally to form a polarized dendritic field and is sensitive to directional movement, color-opponent responses, and orientation-selective response [3,33,39,40]. The second subtype of J-RGCs has a symmetric dendritic field, and their function is not well characterized [3]. Further, YFP is expressed in approximately 12 morphological types of RGCs in Thy1-YFP mice [41]. Altogether, these transgenic mice provide a total of 8 RGC types individually or in small groups, including 1 DS-RGCs, 2 W3-RGCs, 3 αRGCs, 2 J-RGCs, and a mouse strain for a group of 12 types of RGCs.

### 4.2. RGC Death in Optic Neuropathy and Glutamate Excitotoxicity

RGC death is a crucial element in the pathogenesis of many blinding eye diseases, such as optic nerve injury and glaucoma. Recent studies have provided valuable information regarding the molecular mechanisms of RGC death in retinal diseases and suggested multiple mechanistic pathogenesis processes. Although the precise mechanisms of RGC death in TON have not been elucidated, the pathogenesis appears to be multifactorial, and several mechanisms of RGC death have been postulated, such as axonal transport failure, neurotrophic factor deprivation, activation of apoptotic signals, mitochondrial dysfunction, excitotoxic damage, oxidative stress, misbehaving reactive glia and loss of synaptic connectivity [26,27,28,29].

On the other hand, glutamate excitotoxicity has been reported to participate in RGC death by both glaucoma and TON [20,22,69]. Glutamate excitotoxicity is the pathological process by which neurons are damaged and killed by excessive stimulation of glutamate receptors, such as the *N*-methyl-D-aspartate (NMDA) receptor. Excessive stimulation of NMDA receptors can cause excitotoxicity by allowing high levels of calcium ions (Ca^2+^) to enter into cells [70]. Ca^2+^ influx into cells activates a number of enzymes, including phospholipases, endonucleases, and proteases. These enzymes can damage cell structures such as the cytoskeleton, cell membrane, and DNA [71,72]. In addition, a calcium influx through NMDA receptors can cause apoptosis through activation of a cAMP response element binding (CREB) protein shut-off [73]. In the retina, NMDARs are expressed by all RGCs [74,75] and NMDA excitotoxicity is thought to cause RGC death in several retinal diseases [20,22,23,26,69].

In ONC models, the NMDA antagonists, memantine and MK-801, protect RGCs from death [76,77]. Further, the AMPA-KA antagonist, DNQX, also protects RGCs after ONC [78]. Therefore, NMDA excitotoxicity seems to participate in RGC death induced by optic nerve injury. However, to what extent NMDA excitotoxicity or direct mechanic crush causes the death of various types of RGCs has not been systematically investigated. The results from this study and our previous study on NMDA excitotoxicity show that the susceptibilities of BD-RGCs and J-RGCs to ONC and NMDA excitotoxicity vary dramatically. This difference could not be interpreted as showing that ONC is a weaker stressor because the overall survival rates of RGCs labeled by the anti-RBPMS antibody are the same. Therefore, the most plausible interpretation would be that the mechanisms leading to death of these RGC types in ONC and NMDA excitotoxicity might be different. However, this possibility needs to be further tested at a molecular level.

One might argue that the loss of YFP-expressing RGCs in our study might due to loss of YFP expression in RGCs under severe injuries. We have previously examined this possibility by injecting NMDA into the eyes of Thy1-GFP mice, in which most, if not all, RGCs are GFP expressing. We labeled these NMDA-treated retinas using an anti-CASP3 antibody to identify cells undergoing apoptosis. We show that CASP3-positive RGCs are still GFP positive, indicating that RGCs actively undergoing apoptosis are still GFP positive [31]. Therefore, severe injury to RGCs does not lead to selective loss of Thy1-mediated fluorescent reporter expression. Similarly, one might argue whether the anti-RBPMS antibody labels all RGCs in health and severe injured retinas. Several reports showed that almost 100% of RGCs retrogradely labeled by FluoroGold (FG) were also labeled by anti-RBPMS antibody, and approximately 94% to 97% of RBPMS-positive cells were also positive for Thy-1, neurofilament H, and III β-tubulin [49]. In B6.Cg-Tg(Thy1-CFP)23Jrs/J mice, in which Thy1-CFP fluorescence is predominantly expressed in RGCs, RBPMS immunoreactivity is localized to CFP-fluorescent RGCs [51]. In addition, all Brn3a, SMI-32 and melanopsin immunoreactive RGCs express RBPMS immunoreactivity [51]. In retinas injured by ONC or NMDA excitotoxicity, most RBPMS-positive cells are lost [51] but over 95% of remaining RBPMS-positive cells were FG positive and III β-tubulin–positive [50]. Therefore, it was concluded that the anti-RBPMS antibody is a robust reagent that exclusively identifies RGCs and can reliably be used as an RGC marker for quantitative evaluation of RGC degeneration, regardless of the nature and the location of the primary site of the injury and the extent of neurodegeneration [49,50,51].

### 4.3. RGC Type-Specific Susceptibility to Retinal Diseases

Increasing evidence suggests that RGCs are susceptible to various injuries in an RGC type-dependent manner. For instance, the susceptibility of RGCs to elevated IOP depends on soma size, and RGCs with large somata or big axons are more susceptible to elevated IOP [79,80]. Functionally, OFF RGCs appear to be more susceptible to elevated IOP, with defeated synaptic function and dendritic morphology [81,82]. OFF RGCs also exhibited higher rates of cell death and a more rapid decline in both structural and functional organization compared to ON RGCs [10,11,12], but ON RGCs were more susceptible to elevated IOP than ON–OFF RGCs [13]. Further, the transient OFF αRGCs exhibited a higher rate of cell death, while neither sustained OFF αRGCs nor sustained ON αRGCs have reduced synaptic activity due to elevated IOP [12]. Similar to models with elevated IOP, OFF RGCs were more susceptible than ON RGCs to ONC, and ON-sustained RGCs seem to be more susceptible than ON-transient RGCs in models of optic nerve injury [14]. Among αRGCs, intrinsically photosensitive melanopsin-expressing RGCs (ipRGCs), direction-selective (DS) RGCs, and W3-RGCs, αRGCs seem to be the least susceptible type to ONC [15]. However, αRGCs seem to be a more susceptible RGC type in another report [16]. In animal models of glutamate excitotoxicity, larger RGCs at peripheral retina are more sensitive to kainate excitotoxicity while smaller RGCs at central retina are more sensitive to NMDA excitotoxicity [83], and ipRGCs are resistant to NMDA excitotoxicity [84,85]. These results are consistent with the notion that the susceptibility of RGCs to retinal diseases is RGC type specific and might depend upon the type of pathological insults.

Our recent results on the susceptibility of RGCs to NMDA excitotoxicity [31] and the results presented in this study also support the general conclusion that the susceptibility of RGCs to retinal injury is RGC type specific. However, different from some previous reports, our results do not provide a clear correlation between RGC morphology and susceptibility to ONC or NMDA excitotoxicity. Among the five groups of genetically identified RGCs tested in this study and the recent study of the susceptibility of the same RGC types to NMDA excitotoxicity, their susceptibility seems not to be directly correlated to the size of their soma and dendritic field. For instance, the J-RGCs have a much higher susceptibility to both ONC and NMDA excitotoxicity than αRGCs, which are known to have the biggest size of soma and dendritic field, and W3-RGCs, which are the RGCs with the smallest soma and dendritic field [33]. This notion is opposite to the observations by several previous studies [79,80,83]. It was also reported that OFF RGCs appear to be more vulnerable to elevated IOP and ONC than ON RGCs [10,11,12,14,81,82]. ON RGCs are more susceptible to elevated IOP than ON–OFF RGCs [13]. However, the ON and OFF inputs seem not to play a critical role in NMDA-induced RGC death to BD-RGCs, which are ON–OFF RGCs [33,34], and J-RGCs, which are OFF-RGCs [3,33,39,40]. However, the survival rate of J-RGCs is much lower than that of BD-RGCs after ONC in this study. These results raised a critical question of whether the inconsistent results of RGC susceptibility to various pathological insults reported by different studies are due to RGC type-specific susceptibility, the insult-specific effect, or experimental variations.

Although very few studies have directly compared the susceptibility of the same type of RGCs to different pathological insults, the results from this study and our previous study on the susceptibility of RGCs to NMDA excitotoxicity provided such an opportunity. Because different pathological insults might injure RGCs differently, such as NMDA excitotoxicity elevating intracellular calcium [70,71,72,86,87] while ONC reducing axonal transportation [26,27,28,29], it is highly likely that the underlying molecular mechanisms of RGC death are different. Therefore, the susceptibility of the same RGC types might vary significantly to different kinds of pathological insults. Consistent with this possibility, the susceptibility of four out of the five RGC groups/types tested in these two studies varies dramatically with types of injuries, although the overall RGC death remained at a similar level. This is particularly evident for both BD-RGCs and J-RGCs, which have the highest susceptibility to NMDA excitotoxicity, but both of them have much lower susceptibility to ONC and, especially, the survival rate of BD-RGCs after ONC is the second highest among the five RGC groups/types. Therefore, we conclude that the susceptibility of different types of RGCs is likely to be determined by an interaction between the pathological insults and the cell’s intrinsic response mechanisms. Different types of pathological insults might trigger different intrinsic response mechanisms in different RGC types, which might have different efficacy in the activation of the cell death processes in various types of RGCs. If this is a general rule for type-specific RGC death in retinal diseases, it may not be reliable to predict the pattern of RGC death in one condition based on models of other diseases.

Then, an important question is what are the underlying processes that contribute to these inconsistent observations. At least two critical factors might play significant roles in this RGC type-specific susceptibility: the way RGC types are categorized and the types of pathological insults. RGCs are classified into types based on morphological, functional and genetic properties [1,2,3,4,5,6,7,8,9,88]. Most previous studies of RGC type-specific susceptibility are based on morphological and functional classification [10,11,12,13,14,79,80,81,82,83]. Because these morphologically and functionally classified RGC types are likely to have heterogeneous gene expression profiles and, if the gene expression profiles of RGCs contribute to the type-specific susceptibility, how the RGCs are grouped into types could have a significant influence on the observed susceptibility. Consistent with this possibility, it was reported that the same genes could protect some RGC types but facilitate the death of other RGC types after the same injury [19]. Therefore, a more in-depth understanding of the type-specific susceptibility of RGCs to various pathological insults may provide valuable insights into the molecular mechanisms of RGC death. More importantly, this information could help the development of novel cell type-specific and insult-specific treatment strategies.

## 5. Conclusions

In conclusion, we compared the susceptibility of five RGC groups/types to ONC. We showed that BD-RGCs are the most resistant type of RGCs to ONC among the tested RGC types, while he W3-RGCs are the most sensitive cells to ONC. On average, the susceptibilities of both BD-RGCs and αRGCs are significantly lower than the susceptibility of the entire RGC population. The susceptibility of J-RGCs is not different from the susceptibility of the whole RGC population. However, the susceptibility of W3-RGCs is slightly higher than the susceptibility of the entire RGC population. Further, we compared the results of this study to our previous study of the susceptibilities of the same RGC types to NMDA excitotoxicity. We show that BD-RGCs and J-RGCs are more vulnerable to NMDA excitotoxicity than to ONC. These results demonstrate that the susceptibilities of RGC to diseases are not only RGC type specific but also insult dependent.

## Figures and Tables

**Figure 1 cells-09-00677-f001:**
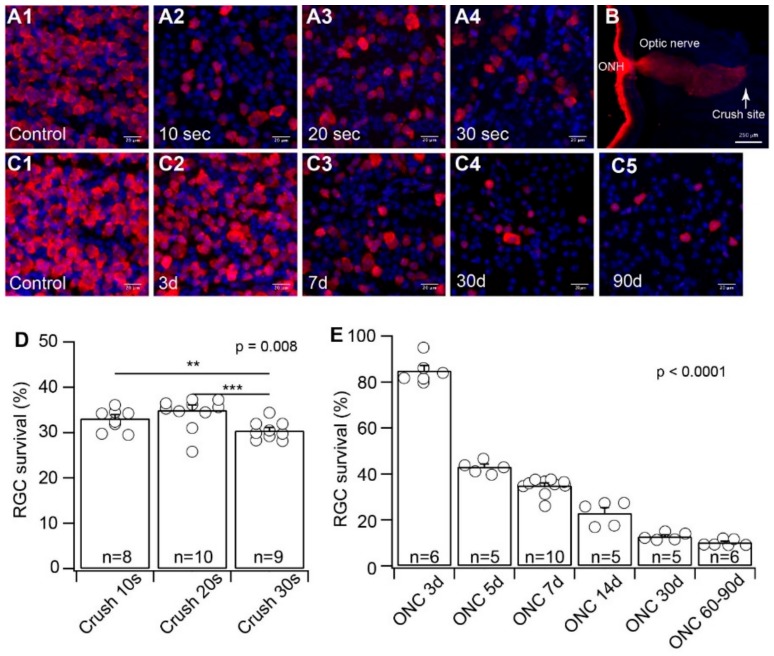
Evaluation of crush time and time after optic nerve crush (ONC) on the death of retinal ganglion cell (RGCs). We first validated whether crushing the optic nerve for different lengths of time causes RGC death to a different extent, and the time course of RGC death after ONC. (**A**) Representative images of flat-mount retinas without injury (**A1**) or seven days after optic nerve clamp for 10 s (**A2**), 20 s (**A3**), and 30 s (**A4**), respectively. RGCs are labeled by anti-RNA-binding protein with multiple splicing (RBPMS) antibody (red), and all nuclei in the ganglion cell layer (GCL) are labeled by DAPI (4′,6-diamidino-2-phenylindole) (blue). (**B**) A representative image of the longitudinal cross-section of the proximal portion of the optic nerve and the posterior portion of the eye seven days after ONC. The posterior retina and proximal optic nerve before the crush site are labeled by Cholera Toxin Subunit B (CTB) (red). The tissue surrounding the optic nerve is labeled by DAPI (blue). The crush site is marked by the CTB labeling. (**C**) Representative images of flat-mount retinas in control (**C1**), three days (**C2**), seven days (**C3**), 30 days (**C4**), and 90 days (**C5**) after 20 s of optic nerve clamp. RGCs are labeled by anti-RBPMS antibody (red), and all nuclei in the GCL are labeled by DAPI (blue). (**D**) Survival rates of RGCs labeled by anti-RBPMS antibody as a function of time of optic nerve clamp (ANOVA test). (**E**) Survival rates of RGCs labeled by anti-RBPMS antibody as a function of time after ONC (ANOVA test). ** 0.001 < *p* < 0.01, *** *p* < 0.001. Each circle indicates an individual eye. The number of n in each column of panels D and E indicates the number of eyes for each group.

**Figure 2 cells-09-00677-f002:**
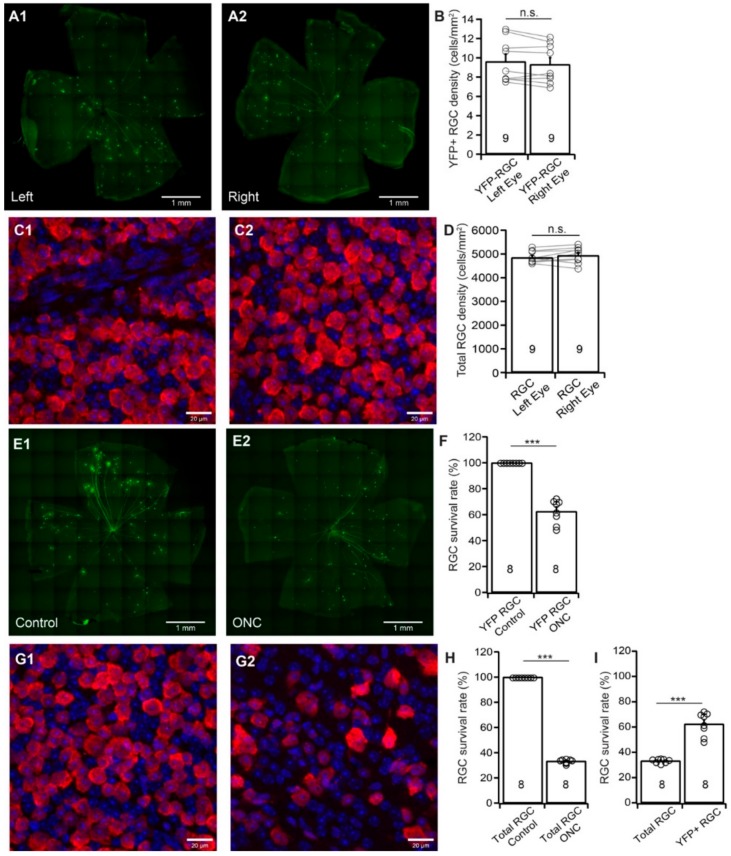
YFP-expressing RGCs of YFP-H mice are more resistant to ONC than total RGCs. We determined whether YFP-expressing RGCs of Thy1-YFP mice could serve as a model to predict overall RGC death in retinal diseases. (**A**) Representative images of flat-mount whole retinas of left (**A1**) and right (**A2**) eyes of a Thy1-YFP mouse. (**B**) Comparison of the densities (cells/mm^2^) of the YFP-expressing RGCs of the left and right eyes of the same group of mice. The densities of YFP-expressing RGCs of left eyes were normalized to the right eyes, and the difference in the densities of YFP-expressing RGCs between the left and right eyes is not statistically significant (paired *t*-test, *p* = 0.118). n.s.: not significant. (**C**) Representative magnified views of flat-mount retinas of left (**C1**) and right (**C2**) eyes of a Thy1-YFP mouse. The RGCs are labeled by an anti-RBPMS antibody (red) and the nuclei in the GCL are labeled by DAPI (blue). (**D**) Comparison of the anti-RBPMS-labeled RGC densities of the left and right eyes of the same group of mice shown in panel B. The densities of anti-RBPMS-labeled RGCs of the left eyes were normalized to the right eyes. The difference between the densities of RGCs in left and right eyes is not statistically significant (paired *t*-test, *p* = 0.252). n.s.: not significant. **E**: Representative images of flat-mount whole retinas of a Thy1-YFP mouse without ONC (E1) and seven days after ONC (E2). (**F**) Comparison of the density of YFP-expressing RGCs of the left eyes seven days after ONC to the uncrushed right eyes of the same mice. The densities of YFP-expressing RGCs of eyes with ONC were normalized to the control eyes. The difference between the control eyes and the eyes with ONC is statistically significant (paired *t*-test, *p* = 0.0003). *** *p* < 0.001. (**G**) Representative magnified views of anti-RBPMS-labeled (red) flat-mount retinas of a Thy1-YFP mouse without ONC (G1) and seven days after ONC (G2). The nuclei in the GCL are labeled by DAPI (blue). (**H**) Comparison of survival rates of anti-RBPMS-labeled RGCs of the eyes seven days after ONC to the control eyes of the same mice. The densities of anti-RBPMS-labeled RGCs of the eyes with ONC were normalized to the control eyes. The difference between the control eyes and the eyes with ONC is statistically significant (paired *t*-test, *p* < 0.0001). *** *p* <0.001. (**I**) Comparison of the survival rates of anti-RBPMS-labeled RGCs and YFP-expressing RGCs from the same eyes seven days after ONC. The densities of anti-RBPMS-labeled and YFP-expressing RGCs of the eyes with ONC were normalized to their uncrushed control eyes of each mouse. The survival rate of YFP-expressing RGCs is significantly higher than that of anti-RBPMS-labeled RGCs from the same eyes (paired *t*-test, *p* < 0.0001). *** *p* < 0.001. The number of n in each column of panels B, D, F, H, and I indicates the number of eyes for each group, and the circles are individual results.

**Figure 3 cells-09-00677-f003:**
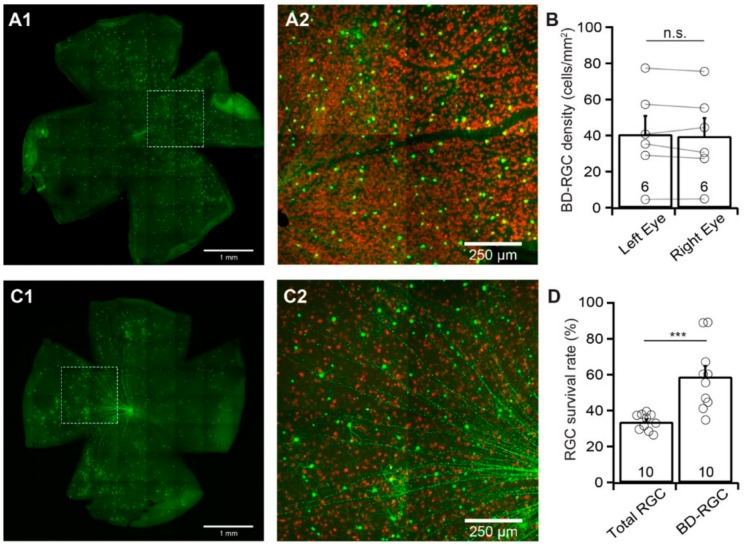
The vulnerability of BD-RGCs to ONC. To determine the susceptibility of BD-RGCs to ONC, we clamped the optic nerve of the left eyes of BD:YFP mice. We quantified the density of YFP-expressing BD-RGCs of the left eyes seven days after ONC and compared results with the density of YFP-expressing BD-RGCs of the uninjured right eyes of the same group of BD:YFP mice. (**A**) A representative image of a flat-mount BD:YFP mouse retina without ONC (**A1**) only showing YFP-expressing BD-RGCs but not anti-RBPMS-labeled RGCs, and a magnified view of the dash-line box of A1 to show the anti-RBPMS staining of all RGCs (red) and YFP-expressing BD-RGCs (green) (**A2**). (**B**) Comparison of the average densities of YFP-expressing BD-RGCs of uncrushed right and left eyes of the same group of BD:YFP mice. The difference between the right and left eyes is not statistically significant (number of mice *n* = 6, paired *t*-test, *p* = 0.4). The circles are individual eyes. n.s.: not significant. (**C**) A representative image from the retina of a BD:YFP mouse with the labeling of YFP-expressing BD-RGCs (green) 7 days after ONC (**C1**) and a magnified view of the boxed area in C1 with labeling of both YFP-expressing BD-RGCs (green) and RGCs labeled by anti-RPBMS antibody (red, **C2**). (**D**) Comparison of the average survival rates of anti-RBPMS-labeled RGCs and YFP-expressing BD-RGCs from the same eyes seven days after ONC. The densities of anti-RBPMS-labeled and YFP-expressing BD-RGCs of the eyes with ONC were normalized to their uncrushed control eyes for each mouse. The survival rates of YFP-expressing BD-RGCs are significantly higher than that of anti-RBPMS-labeled RGCs from the same eyes (number of mice *n* = 10, paired *t*-test, *p* = 0.0005). *** *p* < 0.001. The circles are individual eyes.

**Figure 4 cells-09-00677-f004:**
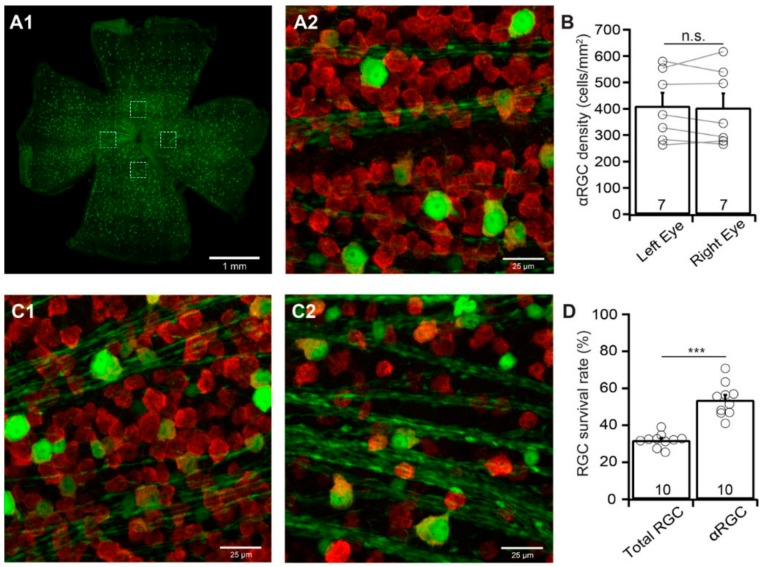
**The susceptibility of αRGCs to ONC.** The density of YFP-expressing αRGCs with ONC on left eyes was quantified, normalized to that of uncrushed right eyes, and compared with the density of anti-RBPMS-labeled RGCs of the same eyes. (**A**) A representative image of a flat-mount Kcng4^Cre^:YFP mouse retina without ONC (**A1**, only showing YFP-expressing αRGCs but not anti-RBPMS staining RGCs)**,** and a magnified view of the left dash-line box of A1 to show the anti-RBPMS-labeled RGCs (red) and YFP-expressing αRGCs (green) (**A2**). (**B**) Comparison of the average densities of YFP-expressing αRGCs of uncrushed right and left eyes of the same group of Kcng4^Cre^:YFP mice. The difference between the right and left eyes is not statistically significant (number of mice *n* = 7, paired *t*-test, *p* = 0.6395). n.s.: not significant. The circles are individual eyes. (**C**) Representative image from the retina of a Kcng4^Cre^:YFP mouse with the labeling of YFP-expressing αRGCs (green) and anti-RBPMS-labeled RGCs (red) without ONC (**C1**) and seven days after ONC (**C2**). (**D**) Comparison of the average survival rates of anti-RBPMS-labeled RGCs and YFP-expressing αRGCs from the same eyes seven days after ONC. The densities of anti-RBPMS-labeled RGCs and YFP-expressing αRGCs of the eyes with ONC were normalized to their uninjured control eyes for each mouse. The survival rate of αRGCs is significantly higher than that of anti-RBPMS-labeled RGCs from the same eyes (number of mice *n* = 10, paired *t*-test, *p* < 0.0001); *** *p* < 0.001.

**Figure 5 cells-09-00677-f005:**
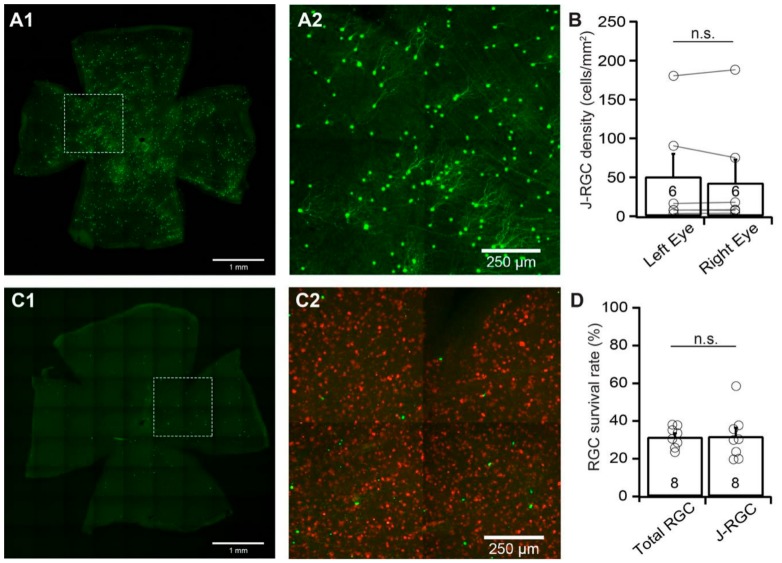
**The vulnerability of J-RGCs to ONC.** The densities of YFP-expressing J-RGCs of left and right eyes of JamB:YFP mice without ONC were compared to determine the YFP expression level of the left and right eyes without injury. The density of the YFP-expressing J-RGCs of left eyes with ONC was quantified, normalized to that of uncrushed right eyes, and compared with the density of anti-RBPMS-labeled RGCs of the same eyes. (A) A representative image of a flat-mount JamB:YFP mouse retina without ONC (**A1**) to show the distribution of YFP-expressing J-RGCs in the retina, and a magnified view of the dash-line box of A1 to show the dendritic morphology of YFP-expressing J-RGCs (**A2**). (**B**) Comparison of the average densities of YFP-expressing J-RGCs from the retinas of the uncrushed right and left eyes of the same group of JamB:YFP mice. The difference between the right and left eyes is not statistically significant (number of mice *n* = 6, paired *t*-test, *p* = 0.7871). n.s.: not significant. The circles are individual eyes. (**C**) A representative image from the retina of a JamB:YFP mouse with the labeling of YFP-expressing J-RGCs (green) 7 days after ONC (**C1**) and a magnified view of the boxed area in C1 with labeling of both YFP-expressing J-RGCs (green) and RGCs labeled by anti-RPBMS antibody (red, **C2**). (**D**) Comparison of the average survival rates of anti-RBPMS-labeled RGCs and YFP-expressing J-RGCs from the same eyes seven days after ONC. The densities of anti-RBPMS-labeled RGCs and YFP-expressing J-RGCs of the eyes with ONC are normalized to their uncrushed control eyes for each mouse. The survival rates of YFP-expressing J-RGCs are not significantly different from that of anti-RBPMS-labeled RGCs from the same eyes (number of mice *n* = 8, paired *t*-test, *p* = 0.9421). n.s.: not significant. The number in each column of panels B and D is n, and the circles are individual results.

**Figure 6 cells-09-00677-f006:**
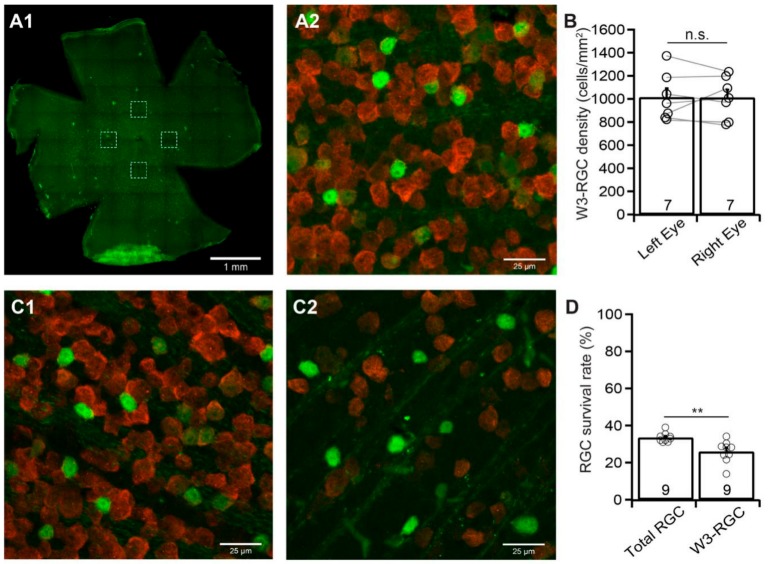
**The susceptibility of W3 RGCs to ONC.** Similar to other RGC types, the densities of YFP-expressing W3-RGCs of left and right eyes of TYW3 mice without ONC are compared to determine the YFP expression level of the left and right eyes without injury. The density of the YFP-expressing W3-RGCs of left eyes with ONC was quantified, normalized to that of uncrushed right eyes, and compared with the density of anti-RBPMS-labeled RGCs of the same eyes. (**A**) A representative image of a flat-mount TYW3 mouse retina without ONC (**A1**, only showing YFP-expressing W3-RGCs but not anti-RBPMS-labeled RGCs)**,** and a magnified view of the left dash-line box of A1 to show the anti-RBPMS-labeled RGCs (red) and YFP-expressing W3-RGCs (green) (**A2**). (**B**) Comparison of the average densities of YFP-expressing W3-RGCs from the retinas of the uncrushed right and left eyes of the same group of TYW3 mice. The difference between the right and left eyes is not statistically significant (number of mice *n* = 7, paired *t*-test, *p* = 0.9561). n.s.: not significant. The circles are individual eyes. (**C**) A representative image from the retina of a TYW3 mouse with the labeling of YFP-expressing W3-RGCs (green) and anti-RBPMS-labeled RGCs (red) without ONC (**C1**) and seven days after ONC (**C2**). (**D**) Comparison of the average survival rates of anti-RBPMS-labeled RGCs and YFP-expressing W3-RGCs from the same eyes seven days after ONC. The densities of anti-RBPMS-labeled RGCs and YFP-expressing W3-RGCs of the eyes with ONC are normalized to their uncrushed control eyes for each mouse. The survival rates of YFP-expressing W3-RGCs are significantly lower than that of anti-RBPMS-labeled RGCs from the same eyes (number of mice *n* = 9, paired *t*-test, *p* = 0.0014). The number in each column of panels B and D is n, and the circles are individual results. ** 0.001 < *p* < 0.01.

**Figure 7 cells-09-00677-f007:**
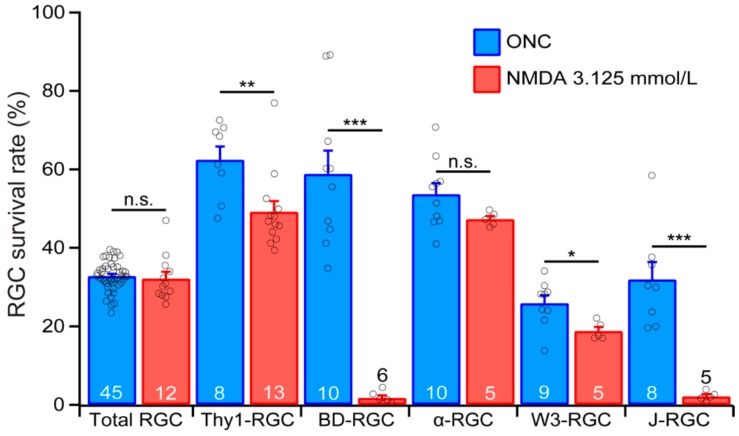
**RGC survivability varies depending on the type of injury.** A comparison of the survival rates of Thy1-RGCs, BD-RGCs, αRGCs, W3-RGCs, and J-RGCs after either ONC (blue) or NMDA excitotoxicity (red). Both insults induced a similar loss of total RGC labeled by the anti-RBPMS antibody. However, their effects on different RGC types are drastically different. The *t*-test results of survival rates of each RGC type/group after ONC or NMDA excitotoxicity are the following: total RGC, *p* = 0.6478; Thy1-RGC, *p* = 0.0066, BD-RGC, *p* < 0.0001; α-RGC, *p* = 0.1419; W3-RGC, *p* = 0.0253; J-RGC, *p* = 0.0003. The number in each column is n, and the circles are results of individual eyes. n.s.: not significant. * 0.01< *p* < 0.05. ** 0.001 < *p* < 0.01. *** *p* < 0.001.

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
