# Peer review of "The Susceptibility of Retinal Ganglion Cells to Optic Nerve Injury is Type Specific"

_cells, 2020, doi:10.3390/cells9030677_

Round 1

Reviewer 1 Report

The manuscript by Yang et al. aims to investigate the selective vulnerability of RGCs to optic nerve crush. They first test the effects of clamping the ON for different durations in order to assess any variation caused by the different time periods. RGC death is then monitored over time, showing consistent results with previous studies. The major data presented provides clear evidence for differences in susceptibility of four different cell types in comparison to the overall RGC population. This is particular evident in the graph presented in Figure 7.

Overall, the manuscript provides evidence for the susceptibility of RGCs to injury is cell-type specific and therefore will be of interest to those working in the areas of cell death and diseases associated with RGC death.

I support the publication of this manuscript, following incorporation of my minor corrections listed here below.

The sex of mice used in the study needs to be clearly stated. Figure 1 panels A3 and A4 state 20 msec and 30 msec, but I assume these should be in sec, rather than msec. Section 4.1 provides a overview of each RGC subtype tested in the study, but a shortened version of this needs to be included in the intro or results section to provide context to the reader. A clear statement of rational for analysing these four subtypes also needs to be included in the intro or results. It would be nice to see a clear statement in the discussion in regards to the authors hypothesis about what causes the differences in susceptibility.

Author Response

Reviewer 1:

Comment: The sex of mice used in the study needs to be clearly stated.

Response: The sentence "Mice in both sexes were used in this study." was added in section 2.1., Animals based on this comment.

Comment: Figure 1 panels A3 and A4 state 20 msec and 30 msec, but I assume these should be in sec, rather than msec.

Response: We thank the reviewer for this comment and corrected this mistake.

Comment: Section 4.1 provides an overview of each RGC subtype tested in the study, but a shortened version of this needs to be included in the intro or results section to provide context to the reader.

Response: Based on the reviewer’s comment, we added the following paragraph into the section of Introduction.

We previously showed that the susceptibility of twenty RGC types to NMDA excitotoxicity varies significantly [33]. One of these mouse lines expresses YFP in a direction-selective RGC type (BD-RGC). The BD-RGCs are a type of ON-OFF direction-selective RGCs (DS-RGCs). In mouse retinas, there are three types of ON-OFF DS-RGCs, tuned to motion in ventral, dorsal, nasal, and temporal. BD-RGCs are sensitive to ventral motion [34,35]. The second mouse line expresses YFP in W3-RGCs. W3-RGCs are the smallest RGCs in the size of the dendritic field and the most numerous RGCs [34]. There are at least two subtypes of W3-RGCs. W3B RGCs are ON-OFF motion-sensitive RGCs [34], and W3D RGCs remain physiologically uncharacterized [36,37]. Both W3-RGC subtypes express YFP in this mouse line. The third mouse line expresses YFP in αRGCs. There are at least three subtypes of αRGCs in mouse retinas, in which some are ON cells and some are OFF cells [38,39]. In this mouse line, YFP is expressed in all three subtypes of αRGCs [15]. The fourth line of the transgenic mice expresses YFP in J-RGCs. There are three subtypes of J-RGCs in the mouse retina, which differ in dendritic tree morphology and possibly visual function [3,7,34]. This mouse line expresses YFP in two subtypes of J-RGCs. One subtype of J-RGCs orients its dendrites toward ventrally to form a polarized dendritic field and is sensitive to directional movement, color-opponent responses, and orientation-selective response [3,34,40,41]. The second subtype of J-RGCs has a symmetric dendritic field, and the function of them is not well characterized [3]. The final mouse line expresses YFP in 12 morphological types of RGCs [42]. Because it has been widely reported that the susceptibility of RGCs varies based on their dendritic ramification patterns, type of their light responses, and their genetic profiles [7,13–15,17,18,43–46], we used these transgenic mouse lines that provided multiple RGC types with various morphological, physiological and genetic profiles.

Comment: A clear statement of rational for analyzing these four subtypes also needs to be included in the intro or results.

Response: As mentioned in response to comment 3, we added the following statement in the introduction:

Because it has been widely reported that the susceptibility of RGCs varies based on their dendritic ramification patterns, type of light their responses, and their genetic profiles [7,9-15,17,18], we used these transgenic mouse lines that provided multiple RGC types with various morphological, physiological and genetic profiles.

Comment: It would be nice to see a clear statement in the discussion in regards to the authors hypothesis about what causes the differences in susceptibility.

Response: We would like to suggest that the underlying molecular mechanisms leading to RGC death not only vary among different RGC types but also among various pathological insults. Although many molecular mechanisms have been postulated for RGC death in glaucoma, TON, glutamatergic excitotoxicity, and diabetic retinopathy, the exact molecular mechanisms that lead RGC death under these disease conditions remain unknown. It is also not clear what causes the differences in susceptibility of different RGC types to these diseases. In this study, our results show a dramatic difference in the survival rates of different RGC types to ONC and NMDA excitotoxicity under the condition that the overall survival rates of RGCs labeled by the anti-RBPMS antibody are the same. This difference could not be interpreted as that ONC is a stronger or weaker stressor than NMDA excitotoxicity, and these two insults share the same molecular mechanisms to lead RGC death. Instead, the most plausible interpretation would be that four out of five groups/types of RGCs respond to ONC and NMDA excitotoxicity differently due to the difference in the underlying molecular mechanisms that lead to RGC death. However, this possibility needs to be further tested at a molecular level. Accordingly, we revised the discussion to clarify this point further.

Reviewer 2 Report

The paper by Dr. Tian and colleagues follows a similar research published recently, which investigates the susceptibility of the same RGC types to NMDA excitotoxicity. The data presented here, together with those of the previous study, concur to demonstrate that the susceptibility of a certain RGC type to a stress not only is type-specific, but is also injury-specific. These observations may contribute to a better understanding and treatment options for traumatic injury of the central nervous system.

My comments:

Line 61: A brief characterization of the four RGC types, in terms of morphology and physiology, would be useful (also in the form of a table). Probably the information given in the discussion paragraph 4.1 should be moved here. Indeed, the characteristics of these RGCs are not properly matter of discussion. In addition, as it appears below, the YFP-expressing RGCs in Thy1-YFP mice include as much as 12 different RGC types. Therefore, it seems inappropriate to declare only four different RGC types analyzed in this study.

Fig 1A3 and 1A4: I think it should be “sec” instead of “msec”.

Lines 223-234: This paragraph refers to a study published previously and does not fit with the result section nor with the title of paragraph 3.2. Some parts may go to the introduction (the characterization of the YFP labeled cells in these retinas) or to the discussion.

Lines 235-249: This is a repetition of the data reported in a previous publication of the same Authors [33] and it should be removed.

Line 517: I think it is J-RGCs instead of W3-RGCs (at least judging from the histograms of fig. 7).

Line 523 “slightly higher”: 31.0 vs 2.1 is not just "slightly" higher.

Lines 527-531: As also noted below by the Authors, these data show that where differences are significant, the survival rate after ONC is always greater than that after NMDA, suggesting that for all these RGC types ONC is a more damaging stress than NMDA. Is there any room for discussion?

Lines 539-540 “of BD-RGCs and J-RGCs”: also W3-RGCs.

Line 568 “(TON)”: This abbreviation has been used before (line 30).

Paragraph 4.2: Most of this discussion is not related to the findings of this study, therefore I suggest to shorten paragraph 4.2.

Lines 604-606 “These results demonstrate that the death of these two RGC types by ONC and NMDA excitotoxicity are unlikely sharing that same mechanism.”: It is quite difficult to hypothesize that the mechanisms leading to cell damage in ONC and in NMDA excitotoxicity are different in these RGC types. They are likely to include oxidative stress and activation of apoptotic signals, which may contribute to cell death both after ONC and after NMDA. As observed above, could it be that for some reason ONC is just a stronger stressor to these cells than NMDA?

Line 251 “the underlying molecular mechanisms of RGC death are different”: This is a possibility, but the available data are not sufficient to give it strength (see also previous comments).

Author Response

Reviewer 2:

Comment: Line 61: A brief characterization of the four RGC types, in terms of morphology and physiology, would be useful (also in the form of a table). Probably the information given in the discussion paragraph 4.1 should be moved here.

Response: Similar to Comment 3 of Reviewer 1, we have provided an overview of each RGC subtype tested in the study, as suggested by the reviewer.

Comment: Indeed, the characteristics of these RGCs are not properly matter of discussion. In addition, as it appears below, the YFP-expressing RGCs in Thy1-YFP mice include as much as 12 different RGC types. Therefore, it seems inappropriate to declare only four different RGC types analyzed in this study.

Response: This concern has been addressed accordingly in this revision.

Comment: Fig 1A3 and 1A4: I think it should be “sec” instead of “msec”.

Response: We thank the reviewer for this comment and corrected this mistake.

Comment: Lines 223-234: This paragraph refers to a study published previously and does not fit with the result section nor with the title of paragraph 3.2. Some parts may go to the introduction (the characterization of the YFP labeled cells in these retinas) or to the discussion.

Response: This paragraph is significantly shortened and combined with the next paragraph as a rationale for the experiment presented in this section.

Comment: Lines 235-249: This is a repetition of the data reported in a previous publication of the same Authors [33] and it should be removed.

Response: We respectively disagree with the reviewer on this point. This is NOT “a repetition of the data reported in a previous publication of the same Authors [33]”. In our previous study indicated by the reviewer [33], we characterized the survival rates of various RGC types to NMDA excitotoxicity. However, the purpose of the current study is focused on the survival rates of different RGC types after ONC. One of the critical elements of these two parallel studies using the same transgenic mouse lines is to test the hypothesis whether the same RGC types respond to different pathological insults, such as NMDA excitotoxicity and ONC, differently. Our results demonstrated this possibility (Figure 7). We believe this is a significant finding and has not been reported previously by anyone else. Therefore, we understand the results presented in this section should be kept for the cross-comparison with our previous study. One might argue that the data shown in Figure 2A and 2B in this manuscript are similar to Figure 1A, 1B and 1C in our previous publication. Both of these data served as controls for NMDA excitotoxicity (our previous publication) and ONC (current study). They are from the same mouse line but different animals. The conclusion from these two animal groups is the same. However, we could not share a control group from a previous study with this study based on a common rule of statistical analysis (each study group should have its control), although they use the same type of animals and have the same conclusion.

Comment: Line 517: I think it is J-RGCs instead of W3-RGCs (at least judging from the histograms of fig. 7).

Response: We agree with the reviewer on this point and corrected this mistake.

Comment: Line 523 “slightly higher”: 31.0 vs 2.1 is not just "slightly" higher.

Response: We replaced the phrase “slightly higher” by the phrase “significantly higher”.

Comment: Lines 527-531: As also noted below by the Authors, these data show that where differences are significant, the survival rate after ONC is always greater than that after NMDA, suggesting that for all these RGC types ONC is a more damaging stress than NMDA. Is there any room for discussion?

Response: First, at seven days after ONC, the survival rate of RGCs labeled by the anti-RBPMS antibody is not significantly different from that of anti-RBPMS labeling of retinas one day after NMDA excitotoxicity. Therefore, we conclude that the overall RGC death under these two conditions (one day after NMDA injection and seven days after ONC) are the same. However, four out of five groups/types of RGCs have higher survival rates seven days after ONC than one day after NMDA injection. These results demonstrate that ONC seems to be less, but not more, damaging stress than NMDA excitotoxicity to these four groups/types of RGCs. It also implies that ONC has to be more damaging stress than NMDA excitotoxicity to other RGC types to maintain a similar level of overall RGC death.

Accordingly, we added the following paragraph into the discussion in this revision.

Furthermore, four out of five groups/types of RGCs have higher survival rates seven days after ONC than one day after NMDA injection, although the survival rates of RGCs labeled by anti-RBPMS antibody are not significantly different under these two conditions. These results indicate that ONC seems to be less damaging stress than NMDA excitotoxicity to these four groups/types of RGCs. These results also imply that ONC has to be more damaging stress than NMDA excitotoxicity to other RGC types to maintain a similar level of overall RGC death.

Comment: Lines 539-540 “of BD-RGCs and J-RGCs”: also W3- RGCs.

Response: We replaced the phrase “of BD-RGCs and J-RGCs” by the phrase “of BD-RGCs, J-RGCs and W3- RGCs”.

Comment: Line 568 “(TON)”: This abbreviation has been used before (line 30).

Response: Corrected.

Comment: Paragraph 4.2: Most of this discussion is not related to the findings of this study, therefore I suggest to shorten paragraph 4.2.

Response: Section 4.2 has been extensively revised and shortened based on this comment.

Comment: Lines 604-606 “These results demonstrate that the death of these two RGC types by ONC and NMDA excitotoxicity are unlikely sharing that same mechanism.”: It is quite difficult to hypothesize that the mechanisms leading to cell damage in ONC and in NMDA excitotoxicity are different in these RGC types. They are likely to include oxidative stress and activation of apoptotic signals, which may contribute to cell death both after ONC and after NMDA. As observed above, could it be that for some reason ONC is just a stronger stressor to these cells than NMDA?

Response: Clearly, the precise mechanisms of RGC death in ONC and NMDA excitotoxicity have not been elucidated. Many possible molecular mechanisms have been postulated. These include axonal transport failure, neurotrophic factor deprivation, activation of apoptotic signals, mitochondrial dysfunction, excitotoxic damage, oxidative stress, misbehaving reactive glia and loss of synaptic connectivity for ONC induced RGC death, and calcium-dependent intracellular enzyme activation, CREB-dependent activation of apoptosis, and destruction of cytoskeleton, cell membrane, and DNA for RGC death induced by NMDA excitotoxicity. The dramatic difference in the survival rates of different RGC types to ONC and NMDA excitotoxicity, such as the BD-RGCs and J-RGCs, demonstrated that the susceptibility of these RGC types to ONC and NMDA excitotoxicity is different. This difference could not be interpreted as that ONC is a stronger or weaker stressor because the overall survival rates of RGCs labeled by the anti-RBPMS antibody are the same. Therefore, the most plausible interpretation would be that the mechanisms leading to the death of these RGC types in ONC and NMDA excitotoxicity might be different. However, this possibility needs to be further tested at a molecular level. We also clarified this point in this revision.

Comment: Line 651 “the underlying molecular mechanisms of RGC death are different”: This is a possibility, but the available data are not sufficient to give it strength (see also previous comments).

Response: As responded to the previous comment, our results show a dramatic difference in the survival rates of different RGC types to ONC and NMDA excitotoxicity, although the overall survival rates of RGCs labeled by the anti-RBPMS antibody injection are the same. This difference could not be interpreted as that ONC is a stronger or weaker stressor than NMDA excitotoxicity, and these two insults share the same molecular mechanisms to lead RGC death. The most plausible interpretation would be that four out of five groups/types of RGCs respond to ONC and NMDA excitotoxicity differently due to the difference in the underlying molecular mechanisms that lead to RGC death.

Reviewer 3 Report

Prior studies performed by Yang et al found that four types of retinal ganglion cells (RGCs) develop distinct vulnerabilities against N-methyl-D-aspartate (NMDA) excitotoxicity. The current work by the same group now reports on the differential survival of the same types of RGCs (BD-RGC, alpha-RGC, J-RGC and W3-RGC) against optic nerve crush (ONC) in mice with genetically labeled RGCs. They also begin their analyses with the examination of a mouse line (Thy1-mice) with expression of YFP to a restricted subpopulation of RGCs. The survival of labeled RGCs of flat-mount retinas are examined between eyes with and without ONC. The authors conclude that BD-RGCs and W3-RGCs are the most and least resistant RGCs to ONC, respectively, while no differences were found between J-RGCs and the whole RGC population. In comparison to NMDA excitotoxicity, the survival rates of BD-RGCs and J-RGCs after ONC were also significantly higher. Overall, the work appears to be well executed and some conclusions appear to be supported by the observations described. Significant deficiencies in clarity and presentation are highlighted.

Specific points:

  1. To date, multiple Thy1-lines with labeled RGCs have been generated. The YFP-labeled RGCs presented in this work appear scarce and with sparse distribution compared to several other reported lines (Fig 2). The authors report that YFP-expressing RGCs vary significantly between Thy1-YFP mice neither by providing a rationale for such finding nor by explaining how such variation may affect the survival analysis of the RGCs between mice. The stock numbers of this and other lines obtained from Jackson Labs are missing and must be reported. It is also unclear whether all YFP-labeled RGCs of this line co-label with the RBPMS antibody. Finally, the authors must clarify whether RBPMS is a pan-RGC marker or what types of RGCs are labeled by the RBPMS antibody.
  2. The same exact issues noted in point 1 also apply to the lines labeled for BD and J-RGCs, because YFP-expression is claimed to vary significantly between mice of the same line. They begin by stating that for this reason Thy1-YFP mice seem to be an unreliable model for studying overall RGC death. Then, why should the lines with BD and J-labeled RGCs be reliable lines to study the survival of these neurons? It seems that some sort normalization analyses of RGCs need to be performed.
  3. No direct evidence of RGC death by ONC is presented except for the fluorescence and RBPMS labeling of RGCs. This issue needs to be addressed somehow.
  4. The discussion is highly unfocused. Some sections are misplaced and belong to the introduction section while other sections are redundant with the introduction (e.g. lines 546-563, 590-606). Further, many sections in the discussion read like a review article (e.g. lines 546-563, 590-606, 608-626). The authors must focus on establishing functional relationships between what is known and their findings and on explaining and discussing the specific implications of their findings in the context of the unmet issue(s) raised by the introduction. The discussion can be also shortened.

Minor points.

5. The authors often refer to the untreated eye as “opposite” eye. The untreated eye should be named the “contralateral” eye.

6. The acronym "CTB" needs to be clarified.

Author Response

Reviewer 3:

Comment: To date, multiple Thy1-lines with labeled RGCs have been generated. The YFP-labeled RGCs presented in this work appear scarce and with sparse distribution compared to several other reported lines (Fig 2). The authors report that YFP-expressing RGCs vary significantly between Thy1-YFP mice neither by providing a rationale for such finding nor by explaining how such variation may affect the survival analysis of the RGCs between mice. The stock numbers of this and other lines obtained from Jackson Labs are missing and must be reported. It is also unclear whether all YFP-labeled RGCs of this line co-label with the RBPMS antibody. Finally, the authors must clarify whether RBPMS is a pan-RGC marker or what types of RGCs are labeled by the RBPMS antibody.

Response: First, it is well known that the strength of YFP gene expression in the Thy1-YFP mice varies significantly among mice. This is also true for all inducible mouse lines, such as the FSTL4-CreER (BD-CreER) and JamB-CreER mice, because the expressing strength of the transgene depends upon the effective dosage of Tamoxifen. To specifically address this variation, we tested every mouse line used in this study and demonstrated that the variation between two eyes of each mouse is very small although the variation among different animals could be very significant. These testing results are all presented in the manuscript. We clearly stated that we only crush one eye of each animal and use the non-crushed contralateral eye as a control. We combined this treatment approach with a paired t-test to minimize the possible effect of the variation between mice on the survival analysis. This is a "standard" approach in the field to process data under this condition. However, the reviewer seems overlooked our description.

Second, the stock numbers of the Thy1-YFP (Stock No: 003782) and Kcng4Cre (Stock No: 029414) mice from The Jackson Lab are included in this revision.

Third, the anti-RBPMS antibody has been shown to be a specific pan-RGC marker, which labels all RGCs but not any other cells in the retina [54–56]. We further clarified this point in this revision.

Comment: The same exact issues noted in point 1 also apply to the lines labeled for BD and J-RGCs, because YFP-expression is claimed to vary significantly between mice of the same line. They begin by stating that for this reason Thy1-YFP mice seem to be an unreliable model for studying overall RGC death. Then, why should the lines with BD and J-labeled RGCs be reliable lines to study the survival of these neurons? It seems that some sort normalization analyses of RGCs need to be performed.

Response: First, the reviewer seems to misread our conclusion. We did not claim Thy1-YFP mice seem to be an unreliable model for studying overall RGC death because YFP-expression varies significantly between mice. Instead, we claim that Thy1-YFP mice seem to be an unreliable model for studying total RGC death because the average survival rate of YFP-expression RGCs in the Thy1-YFP mice is significantly higher than that of RBPMS-labeled RGCs in the same retinas.

Second, as responding to the previous comment, we only crush one eye of each animal and use the non-crushed contralateral eye as a control. We combined this treatment approach with a paired t-test to minimize the possible effect of the variation between mice on the survival analysis. This is a “standard” approach to process data under this condition. The reviewer seems overlooked in our experimental design.

Comment: No direct evidence of RGC death by ONC is presented except for the fluorescence and RBPMS labeling of RGCs. This issue needs to be addressed somehow.

Response: It is not clear what kind of evidence the reviewer would consider as "direct evidence of RGC death." Both the fluorescence labeling of transgenic mouse lines and RBPMS-labeling of RGCs counts are widely used as direct evidence of RGC death in many disease models in numerous published reports.

Comment: The discussion is highly unfocused. Some sections are misplaced and belong to the introduction section while other sections are redundant with the introduction (e.g. lines 546-563, 590-606). Further, many sections in the discussion read like a review article (e.g. lines 546-563, 590-606, 608-626). The authors must focus on establishing functional relationships between what is known and their findings and on explaining and discussing the specific implications of their findings in the context of the unmet issue(s) raised by the introduction. The discussion can be also shortened.

Response: The discussion has been extensively revised to be more focused and shortened.

Comment: The authors often refer to the untreated eye as “opposite” eye. The untreated eye should be named the “contralateral” eye.

Response: Revised as suggested in this revision.

Comment: The acronym "CTB" needs to be clarified.

Response: Revised as suggested.

Round 2

Reviewer 3 Report

Unfortunately, the authors have not adequately addressed my queries/critiques and they appear to engage in the mischaracterization of my statements.

Specific points.

1.  Point 1 was not adequately addressed. The authors claim that “it is well known that the strength of YFP gene expression in the Thy1-YFP mice varies significantly among mice. This is also true for all inducible mouse lines, such as the FSTL4-CreER (BD-CreER) and JamB-CreER mice, because the expressing strength of the transgene depends upon the effective dosage of Tamoxifen.”

These statements are ambiguous and incorrect. “Strength” is a relative term. Instead, the authors in the manuscript compare in absolute terms fluorescent (e.g., Thy1-YFP) or immunolabeled-expressing and non-expressing RGCs. The authors neither use thresholding approaches of any line to normalize the “strength” of the YFP-signal between mice of the same or different transgenic lines nor do they use dosage-dependent analyses of tamoxifen-driven expression between transgenic lines (where applicable, see point 2). While there may be “significant” variation of constitutive Thy1 or conditional tamoxifen-driven expression of fluorescent-labeled RGCs between mice of a given transgenic line, there are also reports of the lack of such variation in YFP-labeled RGCs driven by Thy1 between genotype-matched mice of other transgenic lines. The authors do not provide support from the literature for the existence (and non-existence) of such variations to any of the lines described or of what is unique to the Thy1-YFP line (YFF-H) they use compared to other Thy1-YFP lines reported in the literature (and other than their own studies; e.g., ref. 33). It is incumbent on the authors to educate the readers of a journal serving a broad audience and to provide unbiased support for their statements about every transgenic line examined rather than attempting to mischaracterize my critique. Comparing approaches (and analyses) of crushed eyes against non-crushed contralateral (“opposite”) eyes was never an issue raised by this reviewer as clearly noted in my introductory statement and minor point 5. Finally, the acronyms used for the designated lines and lack of description thereof do little to help less informed readers to understand the nature of the reporter transgenic lines used by this study.

The problem with the YFF-H line is that it shows a ~80-fold variation in YFP-labeled RGCs between mice (e.g., Fig 2B). As I stated in my previous review, the authors claim that for this reason Thy1-YFP mice seem to be an unreliable model for studying overall (total) RGC death. Yet, Fig. 2F shows a significant decrease in YFP-labeled RGCs between eyes with crushed optic nerves and uncrushed contralateral optic nerves, whereas Fig 2H shows also a decrease of higher magnitude of total RGCs from eyes subjected to similar analyses. In contrast to what is stated by the authors, these data indicate that Thy1-YFP-labeled neurons (of the YFF-H line) are reliable indicators of total RGC survival! Regardless, there are two significant problems with these analyses. First and as I stated in my original review, the YFP-labeled neurons are very scarce and sparse in the YFF-H line (single to low double digit density; Fig. 2B). This has the inescapable effect of skewing the data by causing potential under-representation and over-representation of subtypes of RGCs and thus variation between mice of the same line (and differential vulnerability to injury). The second issue is that the authors must use the same units used in Fig 2 B - density of RGCs (instead of % of survival) – for the analysis of the survival of RGCs to compare and unmask known variations in YFP-RGCs and possibly total RGCs between treated eyes of mice of the YFF-H line. This does not prevent the authors from stating the % decrease in survival of RGC in the results’ section and figure legend. The authors should also remove the term “rate” from the labeling of the graphs. Rates are never measured in this work. It is also puzzling why the authors deleted the statement that Thy1-YFP mice seem to be an unreliable model for studying overall RGC death due to NMDA excitotoxicity.

Lastly, the authors did not address whether all YFP-labeled RGCs of the YFF-H line co-label with the RBPMS antibody. This may be relevant since not all reports appear emphatic in stating that every RGCs immunostain with the RBPMS antibody (e.g., some state “most” RGCs immunostain for RBPMS) and this has the potential of compounding the skewed representation of YFP-labeled subtypes of RGCs in untreated and treated eyes as stated earlier.

2.   Likewise, point 2 was not adequately addressed by the authors. Again, the same issues raised in point 1 (and original critique) apply to the lines labeled for BD and J-RGCs (albeit likely for different reasons) and that exhibit large variations of BD and J-RGC densities. For example, the J-RGC line shows a huge variation by almost 200-fold of J-RGC densities between mice (Fig. 5B)!

Hence, my previous critiques stand. The authors neither provide a rationale for the variation of fluorescent-labeled RGCs between mice of each Thy1-YFP, BD or J-RGC line nor do they explain how such variation may affect the survival analysis of RGCs between mice of the same line.

3.  The authors claim that it is not clear what kind of evidence the reviewer would consider as direct evidence of RGC death. Loss of YFP expression or immunostaining of RGC although indicative of cell loss is not direct evidence of cell death. Severe impairments/injuries to RGCs could transiently lead to similar outcomes (e.g., selective loss of antigenicity or reporter expression). The authors appear to be unaware that of countless reports using direct measures of cell death by a variety of approaches depending on the type(s) of cell death examined (e.g. apoptosis, necrosis, etc). Finally, inferences of this work about cell death mechanisms are speculative as none of the experiments of this work address this topic directly.

4.   The discussion remains highly unfocused. The limited scope and nature of this report does not justify the use of subsections in the discussion, some of which read like a review (and some are redundant with the introduction) and contribute to the poor organization of the discussion. For example, the descriptive section of classification of RGC types does not belong to the discussion. Again, the authors must focus on establishing functional relationships between what is known and their findings and on explaining and discussing the specific implications of their findings in the context of the unmet issue(s) raised by the introduction. The discussion needs to be also shortened.

Author Response

Specific points.

  1. Point 1 was not adequately addressed. The authors claim that “it is well known that the strength of YFP gene expression in the Thy1-YFP mice varies significantly among mice. This is also true for all inducible mouse lines, such as the FSTL4-CreER (BD-CreER) and JamB-CreER mice, because the expressing strength of the transgene depends upon the effective dosage of Tamoxifen.”

Comment 1a: These statements are ambiguous and incorrect. “Strength” is a relative term. Instead, the authors in the manuscript compare in absolute terms fluorescent (e.g., Thy1-YFP) or immunolabeled-expressing and non-expressing RGCs. The authors neither use thresholding approaches of any line to normalize the “strength” of the YFP-signal between mice of the same or different transgenic lines nor do they use dosage-dependent analyses of tamoxifen-driven expression between transgenic lines (where applicable, see point 2).

Resp: It is not clear why the term “strength” is “incorrect” for this case. In fact, we normalized the survival YFP-expression RGCs and RBPMS-labeled RGCs of the crushed eye to the contralateral non-crushed eye of each mouse. The rationale and method of this normalization is clearly described in the manuscript. Therefore, the data for the survival is a relative term between the crushed eye and contralateral control eye and using a “relative” term to describe the relative survival rate seems to be appropriate.

Comment 1b: While there may be “significant” variation of constitutive Thy1 or conditional tamoxifen-driven expression of fluorescent-labeled RGCs between mice of a given transgenic line, there are also reports of the lack of such variation in YFP-labeled RGCs driven by Thy1 between genotype-matched mice of other transgenic lines. The authors do not provide support from the literature for the existence (and non-existence) of such variations to any of the lines described or of what is unique to the Thy1-YFP line (YFF-H) they use compared to other Thy1-YFP lines reported in the literature (and other than their own studies; e.g., ref. 33). It is incumbent on the authors to educate the readers of a journal serving a broad audience and to provide unbiased support for their statements about every transgenic line examined rather than attempting to mischaracterize my critique. Comparing approaches (and analyses) of crushed eyes against non-crushed contralateral (“opposite”) eyes was never an issue raised by this reviewer as clearly noted in my introductory statement and minor point 5. Finally, the acronyms used for the designated lines and lack of description thereof do little to help less informed readers to understand the nature of the reporter transgenic lines used by this study.

Resp: It is true that some Thy1 transgenic lines, such as a Thy1-CFP line, express CFP in all RGCs in addition to some amacrine cells. However, we do not use those lines in this study, and, in our opinion, they are less relevant to this study. The reviewer requested to “provide unbiased support for their statements about every transgenic line examined” by “provide support from the literature for the existence (and non-existence) of such variations to any of the lines described or of what is unique to the Thy1-YFP line (YFF-H) they use compared to other Thy1-YFP lines reported in the literature”. Because five different transgenic mouse lines are used in this study, we have briefly summarized the published information to describe these mouse lines in our manuscript. A full scope review of the literature about these five mouse lines used in this study and mouse lines we do not use in this study can only be done in a full-size review paper. Since this reviewer already criticized "The discussion remains highly unfocused" and read like a "reviewer paper," including a more detailed review of each mouse line "from the literature for the existence (and non-existence) of such variations” would certainly make the discussion more “highly unfocused."

Comment 1c: The problem with the YFF-H line is that it shows a ~80-fold variation in YFP-labeled RGCs between mice (e.g., Fig 2B). As I stated in my previous review, the authors claim that for this reason Thy1-YFP mice seem to be an unreliable model for studying overall (total) RGC death.

Resp: In both our first submission and the revised submission, we did NOT claim Thy1-YFP mice are an unreliable model for studying overall RGC death because YFP-expression varies significantly between mice. Instead, we argue that Thy1-YFP mice are an inaccurate model for studying total RGC death because the average survival rate of YFP-expression RGCs in the Thy1-YFP mice is significantly higher than that of RBPMS-labeled RGCs in the same retinas. We also made this point very clear in our response to comment #2 raised by this reviewer in the previous review. Unfortunately, the reviewer seems not to see the difference between these two different statements.

The following is the quote of the second point of the first-round review by this reviewer. Please note the underlined sentence.

The same exact issues noted in point 1 also apply to the lines labeled for BD and J-RGCs, because YFP expression is claimed to vary significantly between mice of the same line. They begin by stating that for

this reason Thy1-YFP mice seem to be an unreliable model for studying overall RGC death. Then, why

 should the lines with BD and J-labeled RGCs be reliable lines to study the survival of these neurons? It seems that some sort normalization analyses of RGCs need to be performed.

In our response to this comment, we clearly stated that "We did not claim Thy1-YFP mice seem to be an unreliable model for studying overall RGC death because YFP-expression varies significantly between mice. Instead, we claim that Thy1-YFP mice seem to be an unreliable model for studying total RGC death because the average survival rate of YFP-expression RGCs in the Thy1-YFP mice is significantly higher than that of RBPMS-labeled RGCs in the same retinas.

The following is the quote of the statement in our first version of the manuscript.

Our results show that the survival rate of YFP-expressing RGCs is significantly higher than that of anti-RBPMS antibody labeled RGCs. This result suggests that the susceptibility of YFP-expressing RGCs in this mouse line is lower than the susceptibility of total RGCs to NMDA excitotoxicity [33]. Therefore, Thy1-YFP mice seem to be an unreliable model for studying overall RGC death due to NMDA excitotoxicity. (line 227 -232)

These results suggest that the susceptibility of YFP-expressing RGCs in this mouse line is lower than the overall susceptibility of RGCs labeled by the anti-RBPMS antibody. Therefore, Thy1-YFP mice seem to be an unreliable model for studying overall RGC death. (line 295-297)

The following is the quote of the statement in our revised version of the manuscript:

Our results show that the survival rate of YFP-expressing RGCs is significantly higher than that of anti-RBPMS antibody labeled RGCs [33] (line 268-270)

These results suggest that the susceptibility of YFP-expressing RGCs in this mouse line is lower than the overall susceptibility of RGCs labeled by the anti-RBPMS antibody. Therefore, Thy1-YFP mice seem to be an unreliable model for studying overall RGC death. (line 351-353)

In neither of these two versions, we claimed “because YFP expression is claimed to vary significantly between mice of the same line” and “for this reason Thy1-YFP mice seem to be an unreliable model for studying overall RGC death.” Instead, we claim that Thy1-YFP mice seem to be an unreliable model for studying total RGC death because the average survival rate of YFP-expression RGCs in the Thy1-YFP mice is significantly higher than that of RBPMS-labeled RGCs in the same retinas. We believe that these two statements are fundamentally different.  

Comment 1d: Yet, Fig. 2F shows a significant decrease in YFP-labeled RGCs between eyes with crushed optic nerves and uncrushed contralateral optic nerves, whereas Fig 2H shows also a decrease of higher magnitude of total RGCs from eyes subjected to similar analyses. In contrast to what is stated by the authors, these data indicate that Thy1-YFP-labeled neurons (of the YFF-H line) are reliable indicators of total RGC survival!

Resp: We respectively disagree with the reviewer on this point. Although both the numbers of YFP-expression RGCs and anti-RBPMS antibody labeled RGC are significantly decreased after ONC from the same retinas, the magnitudes of the decrease of these two groups are significantly different. This is specifically tested and presented in Figure 2I. This is also the point we repeatedly emphasized. Because the survival rate of YFP-expressing RGCs is significantly higher than that of anti-RBPMS antibody labeled RGCs (Fig 2I), we do not think that Thy1-YFP-labeled neurons (of the YFF-H line) are quantitatively reliable indicators of total RGC survival! Unfortunately, the reviewer either overlooked this point or do not understand this point. Please note, we are looking for quantitative evaluation, not qualitative evaluation!

Comment 1e: Regardless, there are two significant problems with these analyses. First and as I stated in my original review, the YFP-labeled neurons are very scarce and sparse in the YFF-H line (single to low double digit density; Fig. 2B). This has the inescapable effect of skewing the data by causing potential under-representation and over-representation of subtypes of RGCs and thus variation between mice of the same line (and differential vulnerability to injury).

Resp: The comment will be true only if we do not normalize the data. However, we normalized the number of YFP-labeled neurons of the crushed eye to the contralateral (non-crushed) eye of each mouse. This normalization is based on the results that no matter how significant the variation of YFP-labeled neurons between mice, the variation of YFP-labeled neurons between left and right eyes are not significant (Fig 2B). After the normalization, the data is presented as survival rate in Figs 2F, 2H, and 2I. By examining the distribution of data points of individual eyes in these three panels, it is evident that the points are not skewed. This so-called “self-normalization” is a common way to be used to normalize biological data.

Comment 1f: The second issue is that the authors must use the same units used in Fig 2 B - density of RGCs (instead of % of survival) – for the analysis of the survival of RGCs to compare and unmask known variations in YFP-RGCs and possibly total RGCs between treated eyes of mice of the YFF-H line. This does not prevent the authors from stating the % decrease in survival of RGC in the results’ section and figure legend. The authors should also remove the term “rate” from the labeling of the graphs. Rates are never measured in this work. It is also puzzling why the authors deleted the statement that Thy1-YFP mice seem to be an unreliable model for studying overall RGC death due to NMDA excitotoxicity.

Resp: As responded to the previous point (Comment 1e), we normalized the number of YFP-labeled neurons of the crushed eye to the contralateral (non-crushed) eye of each mouse. Once the data is normalized, it is a rate of the crushed eye to the non-crushed contralateral eye of each mouse. Therefore, it cannot be expressed as cell density (cell number/mm2) but a ratio of crushed eye/non-crushed eye x 100.

Comment 1g: Lastly, the authors did not address whether all YFP-labeled RGCs of the YFF-H line co-label with the RBPMS antibody. This may be relevant since not all reports appear emphatic in stating that every RGCs immunostain with the RBPMS antibody (e.g., some state “most” RGCs immunostain for RBPMS) and this has the potential of compounding the skewed representation of YFP-labeled subtypes of RGCs in untreated and treated eyes as stated earlier.

Resp: First, we examined every retina used in this study and confirmed that all YFP-labeled RGCs of the YFF-H line co-label with the RBPMS antibody. Second, it has been reported that “Quantitative analysis showed that almost 100% of RGCs labeled by FG were also RBPMS-positive, irrespective of their location relative to the optic nerve head. Approximately 94% to 97% of RBPMS-positive cells were also positive for Thy-1, neurofilament H, and III b-tubulin” (Kwong et al., IOVS, 2010). “In mouse and rat retina, most RBPMS cells are lost following optic nerve crush or transection at three weeks, and all Brn3a, SMI-32 and melanopsin immunoreactive RGCs also express RBPMS immunoreactivity. RBPMS immunoreactivity is localized to CFP-fluorescent RGCs in the B6.Cg-Tg(Thy1-CFP)23Jrs/J mouse line.” (Rodriguez et al., JCN 2015). In the second reference, it states, "most RBPMS cells are lost following optic nerve crush," but not "most RGCs are labeled." There is no evidence to support the statement that “this has the potential of compounding the skewed representation of YFP-labeled subtypes of RGCs in untreated and treated eyes as stated earlier."

Comment 2.   Likewise, point 2 was not adequately addressed by the authors. Again, the same issues raised in point 1 (and original critique) apply to the lines labeled for BD and J-RGCs (albeit likely for different reasons) and that exhibit large variations of BD and J-RGC densities. For example, the J-RGC line shows a huge variation by almost 200-fold of J-RGC densities between mice (Fig. 5B)!

Hence, my previous critiques stand. The authors neither provide a rationale for the variation of fluorescent-labeled RGCs between mice of each Thy1-YFP, BD, or J-RGC line, nor do they explain how such variation may affect the survival analysis of RGCs between mice of the same line.

Resp: As responded above (Comment 1b), we think neither the previous critiques nor the current critiques stand for this point.

Comment 3.  The authors claim that it is not clear what kind of evidence the reviewer would consider as direct evidence of RGC death. Loss of YFP expression or immunostaining of RGC although indicative of cell loss is not direct evidence of cell death. Severe impairments/injuries to RGCs could transiently lead to similar outcomes (e.g., selective loss of antigenicity or reporter expression). The authors appear to be unaware that of countless reports using direct measures of cell death by a variety of approaches depending on the type(s) of cell death examined (e.g. apoptosis, necrosis, etc). Finally, inferences of this work about cell death mechanisms are speculative as none of the experiments of this work address this topic directly.

Resp: The purpose of this study is to compare the susceptibility of different RGC types. Apparently, the reviewer appears not to realize that none of the methods mentioned in the comment, such as apoptosis or necrosis, will be able to label RGC death in an RGC type (such as BD-RGC or J-RGC) specific manner. It has also been reported that RGC death might not undergo the same mechanism, such as apoptosis. Therefore, the methods suggested by the reviewer will not serve the purpose of this study.

More importantly, we have previously examined the possibility of whether severe impairments/injuries to RGCs transiently lead to selective loss of antigenicity or reporter expression, as the reviewer suggested. We injected NMDA into the eyes of Thy1-GFP mice, in which most, if not all, RGCs are GFP-expressing, and labeled NMDA-treated retinas using an anti-CASP3 antibody to identify cells undergoing apoptosis. The results show that CASP3-positive RGCs are still GFP-positive, indicating that RGCs actively undergoing apoptosis are still GFP positive. These results have been published recently (Christensen et al., Front Neurosci 2019). Therefore, the injury to RGCs by NMDA excitotoxicity does not lead to selective loss of YFP reporter expression.

In terms of the inferences of our work about cell death mechanisms, our results do not pinpoint a mechanism proposed by previous reports. However, in our opinion, the difference in susceptibility of various RGC types to the same injury (ONC in this study), or the difference in susceptibility of the same RGC type (BD-RGC or J-RGC) to different injuries (ONC and NMDA excitotoxicity) strongly suggest that different RGC types might respond to different pathological insults through different mechanisms. We also acknowledged that a more in-depth understanding of the type-specific susceptibility of RGCs to various pathological insults might provide valuable insights into the molecular mechanisms of RGC death.

Comment 4.   The discussion remains highly unfocused. The limited scope and nature of this report does not justify the use of subsections in the discussion, some of which read like a review (and some are redundant with the introduction) and contribute to the poor organization of the discussion. For example, the descriptive section of classification of RGC types does not belong to the discussion. Again, the authors must focus on establishing functional relationships between what is known and their findings and on explaining and discussing the specific implications of their findings in the context of the unmet issue(s) raised by the introduction. The discussion needs to be also shortened.

Resp: This seems to be a contradictory comment to one point (Comment 1b) made in the first comment by this reviewer that states, "It is incumbent on the authors to educate the readers of a journal serving a broad audience and to provide unbiased support for their statements about every transgenic line examined ….” As responded to the Comment 1b, we believe a full scope review of the literature about the five mouse lines used in this study and mouse lines we do not use in this study can only be done in a full-size review paper. Because this reviewer already commented "The discussion remains highly unfocused" and read like a "reviewer paper," including more detailed review of each mouse line "from the literature for the existence (and non-existence) of such variations" would certainly make the discussion more “highly unfocused” and like a review paper.